# REG: Rectified Gradient Guidance for Conditional Diffusion Models

**Zhengqi Gao** [1]  **Kaiwen Zha** [1]  **Tianyuan Zhang** [1]  **Zihui Xue** [2]  **Duane S. Boning** [1]

## Abstract

Guidance techniques are simple yet effective for improving conditional generation in diffusion models. Albeit their empirical success, the practical implementation of guidance diverges significantly from its theoretical motivation. In this paper, we reconcile this discrepancy by replacing the scaled marginal distribution target, which we prove theoretically invalid, with a valid scaled joint distribution objective. Additionally, we show that the established guidance implementations are approximations to the intractable optimal solution under no future foresight constraint. Building on these theoretical insights, we propose rectified gradient guidance (REG), a versatile enhancement designed to boost the performance of existing guidance methods. Experiments on 1D and 2D demonstrate that REG provides a better approximation to the optimal solution than prior guidance techniques, validating the proposed theoretical framework. Extensive experiments on class-conditional ImageNet and text-to-image generation tasks show that incorporating REG consistently improves FID and Inception/CLIP scores across various settings compared to its absence. Our code is publicly available at: https://github.com/zhengqigao/REG/.

## 1. Introduction

Generative machine learning endeavors to model the underlying data distribution, enabling the synthesis of new data samples that closely mirror the characteristics of the original dataset. While many generative model families (Kingma, 2013; Goodfellow et al., 2020) have emerged over time, the recent surge in diffusion models (Ho et al., 2020; Song et al., 2021) has marked a significant breakthrough, allowing for diverse and high-quality generation. These diffusion

models now dominate a wide range of tasks, such as class-conditional image generation (Peebles & Xie, 2023; Karras et al., 2024b), text-to-image generation (Ramesh et al., 2022; Rombach et al., 2022), video generation (Ho et al., 2022), and audio and speech synthesis (Kong et al., 2021). Despite the varied theoretical foundations, such as DDPM (Ho et al., 2020), score matching (Song et al., 2021), Schrödinger Bridge (De Bortoli et al., 2021), and flow matching (Lipman et al., 2023), diffusion models converge on a unified implementation: a forward process progressively corrupts data by adding Gaussian noise, while a reverse process, parameterized by a neural network, is trained to denoise and reconstruct high-quality samples from pure noise.

One pivotal factor behind the success of diffusion models is the guidance technique (Dhariwal & Nichol, 2021; Ho & Salimans, 2022; Kynkäänniemi et al., 2024; Karras et al., 2024a). Concretely, guidance is a sampling-time method that balances mode coverage and sample fidelity (Ho & Salimans, 2022) by updating the noise prediction network output as a weighted sum of its original output and a user-defined guidance signal, with the mixing coefficient (i.e., guidance strength) controlled by hyper-parameters. When initially proposed by Dhariwal & Nichol (2021), the guidance signal is provided by an auxiliary classifier trained alongside the diffusion model, giving rise to the name classifier guidance. Subsequently, Ho & Salimans (2022) eliminate the need for an additional classifier, relying instead on an implicit Bayes posterior classifier to produce the guidance signal. This classifier-free guidance method has since become a pervasive component of modern diffusion models, enabling effective conditional generation across various applications.

Despite the practical effectiveness of the guidance technique, its motivation and theoretical formulation remain poorly understood and, at times, conflicting in existing literature. Specifically, guidance is originally stated as constructing a new noise prediction network post-training corresponding to sampling from a scaled distribution (Ho & Salimans, 2022). However, recent works (Bradley & Nakkiran, 2024; Chidambaram et al., 2024) using Gaussian mixture case studies reveal that this newly constructed noise prediction network does not result in sampling from the intended scaled distribution. This fundamental inconsistency weakens the theoretical foundation of current guidance techniques, raising misinterpretations and concerns about the implemen-

[1]Massachusetts Institute of Technology. [2]University of Texas at Austin. Correspondence to: Zhengqi Gao <zhengqi@mit.edu>.

*Proceedings of the 42^nd International Conference on Machine Learning*, Vancouver, Canada. PMLR 267, 2025. Copyright 2025 by the author(s).

tation optimality. In this paper, we reconcile this conflict and build a unified theoretical framework for understanding guidance techniques in conditional diffusion models. Our main contributions include:

(1) We establish that scaling the marginal distribution at the denoising endpoint, as used in current guidance literature, conflicts with the inherent constraints of the reverse denoising process of diffusion models (§ 3).

(2) We ground our theory in scaling the joint distribution associated with the entire denoising chain, interpreting guidance methods as approximations to the intractable optimal solution with quantified error bounds (§ 4).

(3) Finally, we present rectified gradient guidance (REG), a versatile enhancement compatible with various guidance techniques (§ 4). Comprehensive experiments validate our theoretical framework and justify the effectiveness of the proposed REG method (§ 5).

## 2. Preliminary

We briefly review guidance techniques (Dhariwal & Nichol, 2021; Ho & Salimans, 2022) developed for conditional image generation in diffusion models. For simplicity, we adopt the notation of discrete-time DDPM (Ho et al., 2020) throughout this paper, while noting that all of our results can be similarly derived in other diffusion settings (Song et al., 2021; Karras et al., 2024b). Let us denote the conditional distribution of interest as $q(\mathbf{x}_0|\mathbf{y})$. We attempt to learn a diffusion model to generate samples $\mathbf{x}_0 \in \mathcal{X} \subset \mathbb{R}^D$ given the conditioning variable $\mathbf{y} \in \mathcal{Y}$, where $\mathbf{y}$ can be either continuous (e.g., a text embedding) or discrete (e.g., a class label). Starting from a clean data $\mathbf{x}_0$, the forward process of DDPM produces samples $\{\mathbf{x}_t\}_{t=0}^T$ at progressively higher noise levels by gradually injecting Gaussian noises through $T$-step transitions $q(\mathbf{x}_t|\mathbf{x}_{t-1})$:

$$q(\mathbf{x}_t|\mathbf{x}_{t-1}) = \mathcal{N}(\mathbf{x}_t|\sqrt{\alpha_t}\mathbf{x}_{t-1}, (1-\alpha_t)\mathbf{I}),$$
$$q(\mathbf{x}_{0:T}|\mathbf{y}) = q(\mathbf{x}_0|\mathbf{y})\prod_{t=1}^T q(\mathbf{x}_t|\mathbf{x}_{t-1}), \quad (1)$$

where $\{\alpha_t\}_{t=1}^T$ is a decreasing series in $[0,1]$ controlling the noise variance. The reverse process of DDPM starts from a noisy sample $\mathbf{x}_T$, and gradually denoises it by $T$-step transitions $p_\theta(\mathbf{x}_{t-1}|\mathbf{x}_t, \mathbf{y})$, yielding a clean sample $\mathbf{x}_0$:

$$p_\theta(\mathbf{x}_{t-1}|\mathbf{x}_t, \mathbf{y}) = \mathcal{N}(\mathbf{x}_{t-1}|\boldsymbol{\mu}_{\theta,t}, \sigma_t^2\mathbf{I}),$$
$$p_\theta(\mathbf{x}_{0:T}|\mathbf{y}) = p_\theta(\mathbf{x}_T|\mathbf{y})\prod_{t=1}^T p_\theta(\mathbf{x}_{t-1}|\mathbf{x}_t, \mathbf{y}), \quad (2)$$

where $\theta$ represents learnable parameters, and $p_\theta(\mathbf{x}_T|\mathbf{y})$ is usually fixed to a standard Gaussian distribution regardless of $\mathbf{y}$, i.e., $p_\theta(\mathbf{x}_T|\mathbf{y}) = \mathcal{N}(\mathbf{x}_T|\mathbf{0}, \mathbf{I})$. The variance term $\sigma_t^2$

is fixed in DDPM (Ho et al., 2020), but can also be learned as a function of $(\mathbf{x}_t, t, \mathbf{y})$ (Nichol & Dhariwal, 2021). The mean $\boldsymbol{\mu}_{\theta,t} \in \mathbb{R}^D$ is parameterized by a noise prediction network $\boldsymbol{\epsilon}_\theta(\mathbf{x}_t, t, \mathbf{y}) : \mathcal{X} \times \mathbb{R}^+ \times \mathcal{Y} \to \mathbb{R}^D$ as follows:

$$\boldsymbol{\mu}_{\theta,t} = \frac{1}{\sqrt{\alpha_t}}\left(\mathbf{x}_t - \frac{1-\alpha_t}{\sqrt{1-\bar{\alpha}_t}}\boldsymbol{\epsilon}_\theta(\mathbf{x}_t, t, \mathbf{y})\right), \quad (3)$$

where $\bar{\alpha}_t = \prod_{i=1}^t \alpha_i$. The noise prediction network $\boldsymbol{\epsilon}_\theta(\mathbf{x}_t, t, \mathbf{y})$ is trained by minimizing the weighted L2 norm between its output and the actual noise motivated by variational evidence lower bound (Ho et al., 2020). From now on, unless explicitly stated, we will omit the arguments and use the shorthand notation $\boldsymbol{\epsilon}_{\theta,t} = \boldsymbol{\epsilon}_\theta(\mathbf{x}_t, t, \mathbf{y})$ for brevity.

Applying guidance techniques in conditional diffusion models is achieved by replacing $\boldsymbol{\epsilon}_{\theta,t}$ with a refined version $\bar{\boldsymbol{\epsilon}}_{\theta,t}$ in the reverse denoising process Eq. (3) during sampling. Below we delineate three prominent forms of $\bar{\boldsymbol{\epsilon}}_{\theta,t}$, corresponding to classifier guidance (CG) (Dhariwal & Nichol, 2021), classifier-free guidance (CFG) (Ho & Salimans, 2022), and auto-guidance (AutoG) (Karras et al., 2024a):

$$\begin{aligned} \text{CG:} \quad & \bar{\boldsymbol{\epsilon}}_{\theta,t} = \boldsymbol{\epsilon}_{\theta,t} - w\sqrt{1-\bar{\alpha}_t}\nabla_{\mathbf{x}_t}\log p_\phi(\mathbf{y}|\mathbf{x}_t), \\ \text{CFG:} \quad & \bar{\boldsymbol{\epsilon}}_{\theta,t} = \boldsymbol{\epsilon}_{\theta,t} + w\left(\boldsymbol{\epsilon}_{\theta,t} - \boldsymbol{\epsilon}_\theta(\mathbf{x}_t, t)\right), \quad (4) \\ \text{AutoG:} \quad & \bar{\boldsymbol{\epsilon}}_{\theta,t} = \boldsymbol{\epsilon}_{\theta,t} + w\left(\boldsymbol{\epsilon}_{\theta,t} - \boldsymbol{\epsilon}_{\theta_{\text{bad}}}(\mathbf{x}_t, t, \mathbf{y})\right), \end{aligned}$$

where the hyper-parameter $w \in \mathbb{R}^+$ controls the guidance strength, and the shorthand notations $\bar{\boldsymbol{\epsilon}}_{\theta,t} = \bar{\boldsymbol{\epsilon}}_\theta(\mathbf{x}_t, t, \mathbf{y})$ is used for clarity. We emphasize that Eq. (4) is applied for $t = 1, 2, \cdots, T$, not $t = 0$. Since in our notation, $\boldsymbol{\epsilon}_\theta(\mathbf{a}, 1, \mathbf{y})$ is the noise prediction at the final time step used to update from $\mathbf{x}_1 = \mathbf{a}$ to $\mathbf{x}_0$, and $\boldsymbol{\epsilon}_\theta(\cdot, 0, \cdot)$ is never used in the reverse denoising process and meaningless.

In CG, an auxiliary classifier $p_\phi(\mathbf{y}|\mathbf{x}_t)$ with learnable parameters $\phi$ is trained alongside $\boldsymbol{\epsilon}_{\theta,t}$ to provide the guidance signal during sampling (Dhariwal & Nichol, 2021). In contrast, CFG removes the need for an external classifier by utilizing the diffusion model itself (Ho & Salimans, 2022). The elegance of CFG lies in its training process, where the conditioning variable $\mathbf{y}$ is randomly dropped. This enables a single noise prediction network $\boldsymbol{\epsilon}_{\theta,t}$ to operate either conditionally as $\boldsymbol{\epsilon}_\theta(\mathbf{x}_t, t, \mathbf{y})$, or unconditionally as $\boldsymbol{\epsilon}_\theta(\mathbf{x}_t, t) = \boldsymbol{\epsilon}_\theta(\mathbf{x}_t, t, \mathbf{y} = \varnothing)$ by masking the conditioning variable $\mathbf{y} = \varnothing$ during sampling. Compared to CFG, AutoG further enhances guidance by using a degraded version $\theta_{\text{bad}}$ of the diffusion model, such as a checkpoint from an earlier stage of training (Karras et al., 2024a).

## 3. Guidance Theory Pitfall

**Common Interpretation of Guidance.** The original motivation underlying guidance (which later we will correct) is that we attempt to sample from a constructed $\bar{p}_\theta(\mathbf{x}_0|\mathbf{y})$

given a trained diffusion $p_\theta(\mathbf{x}_0|\mathbf{y})$ (Ho & Salimans, 2022):

$$\bar{p}_\theta(\mathbf{x}_0|\mathbf{y}) \propto p_\theta(\mathbf{x}_0|\mathbf{y}) \cdot R_0(\mathbf{x}_0, \mathbf{y}), \quad (5)$$

where the reward $R_0(\mathbf{x}_0, \mathbf{y}) : \mathcal{X} \times \mathcal{Y} \to \mathbb{R}^+$ encourages sampling $\mathbf{x}_0$ more frequently from where $R_0(\mathbf{x}_0, \mathbf{y})$ is large. The rewards associated with CG, CFG, and AutoG are: [1]

$$\text{CG:} \quad R_0(\mathbf{x}_0, \mathbf{y}) = [p_\phi(\mathbf{y}|\mathbf{x}_0)]^w,$$

$$\text{CFG:} \quad R_0(\mathbf{x}_0, \mathbf{y}) = \left[\frac{p_\theta(\mathbf{x}_0|\mathbf{y})}{p_\theta(\mathbf{x}_0)}\right]^w, \quad (6)$$

$$\text{AutoG:} \quad R_0(\mathbf{x}_0, \mathbf{y}) = \left[\frac{p_\theta(\mathbf{x}_0|\mathbf{y})}{p_{\theta_{\text{bad}}}(\mathbf{x}_0|\mathbf{y})}\right]^w.$$

Intuitively, the CG reward encourages sampling from where $p_\phi(\mathbf{y}|\mathbf{x}_0)$ is large, which is advantageous for conditional generation tasks (e.g., emphasizing class-specific characteristics or aligning better with a text embedding). For the CFG reward, Bayes' theorem allows us to rewrite it as $R_0(\mathbf{x}_0, \mathbf{y}) \propto [p_\theta(\mathbf{y}|\mathbf{x}_0)]^w$, implying that the CFG reward differs from the CG reward merely in its use of an implicit Bayes posterior classifier derived from the diffusion model.

Eqs. (5)-(6) are the commonly believed guidance interpretations. To ensure that the reverse denoising process indeed samples from $\bar{p}_\theta(\mathbf{x}_0|\mathbf{y})$, two key relationships are required. The first is the score function formula (Song et al., 2021; Ho et al., 2020):

$$\nabla_{\mathbf{x}_t} \log p_\theta(\mathbf{x}_t|\mathbf{y}) = -\frac{\boldsymbol{\epsilon}_\theta(\mathbf{x}_t, t, \mathbf{y})}{\sqrt{1 - \bar{\alpha}_t}},$$

$$\nabla_{\mathbf{x}_t} \log p_\theta(\mathbf{x}_t) = -\frac{\boldsymbol{\epsilon}_\theta(\mathbf{x}_t, t)}{\sqrt{1 - \bar{\alpha}_t}}. \quad (7)$$

In the discrete-time DDPM formulation, strictly speaking, Eq. (7) is valid only for time steps $t = 1, 2, \cdots, T$, but not for $t = 0$. In contrast, the continuous-time stochastic differential equation (SDE) formulation of diffusion models (Song et al., 2021; Lu et al., 2022) naturally extends the validity of the equation to all $t \in [0, T]$, including $t = 0$, which we adopt in this paper. See Appendix A for details.

Next, by taking the derivative of the logarithm of Eq. (5), we obtain the second relationship: $\nabla_{\mathbf{x}_0} \log \bar{p}_\theta(\mathbf{x}_0|\mathbf{y}) = \nabla_{\mathbf{x}_0} \log p_\theta(\mathbf{x}_0|\mathbf{y}) + \nabla_{\mathbf{x}_0} \log R_0(\mathbf{x}_0, \mathbf{y})$. It is stated in the literature (Ho & Salimans, 2022) that substituting Eq. (7) into this gradient equation and using the rewards shown in Eq. (6) allows CG, CFG, and AutoG to construct a new

noise prediction network $\bar{\boldsymbol{\epsilon}}_{\theta,t}$, which can then be used to sample from $\bar{p}_\theta$ following the denoising process Eqs. (2)-(3). However, a closer examination reveals that this derivation would yield equations for $\bar{\boldsymbol{\epsilon}}_{\theta,t}$ associated with only $t = 0$. On the contrary, Eq. (4), which has been employed by CG, CFG, and AutoG in practice, applies to all time steps $t = 1, 2, \cdots, T$, but not $t = 0$.

This discrepancy highlights a significant gap between the theoretical derivations and the practical implementations of current guidance techniques. Another way to illustrate this gap is to observe that Eq. (5) implicitly assumes no change to $p_\theta(\mathbf{x}_t|\mathbf{y})$ for $t = 1, 2, \cdots, T$. In contrast, the actual practice Eq. (4) modifies $\boldsymbol{\epsilon}_{\theta,t}$, thereby altering these distributions through the score function formula in Eq. (7).

**Invalid Marginal Scaling.** Since the theoretical interpretation and practical implementations only differ in the time steps at which Eq. (4) is applied, it seems appealing to bridge this gap by re-framing the original goal. Specifically, instead of enforcing a single scaled constraint exclusively on the marginal at the denoising endpoint, we redefine the objective to impose scaled constraints on all marginals (Ho & Salimans, 2022): [2]

$$\bar{p}_\theta(\mathbf{x}_t|\mathbf{y}) \propto p_\theta(\mathbf{x}_t|\mathbf{y}) \cdot R_t(\mathbf{x}_t, \mathbf{y}), \quad (8)$$

where $t = 1, 2, \cdots, T$, and the rewards are:

$$\text{CG:} \quad R_t(\mathbf{x}_t, \mathbf{y}) = [p_{\phi, X_t}(\mathbf{y}|\mathbf{x}_t)]^w,$$

$$\text{CFG:} \quad R_t(\mathbf{x}_t, \mathbf{y}) = \left[\frac{p_{\theta, X_t}(\mathbf{x}_t|\mathbf{y})}{p_{\theta, X_t}(\mathbf{x}_t)}\right]^w, \quad (9)$$

$$\text{AutoG:} \quad R_t(\mathbf{x}_t, \mathbf{y}) = \left[\frac{p_{\theta, X_t}(\mathbf{x}_t|\mathbf{y})}{p_{\theta_{\text{bad}}, X_t}(\mathbf{x}_t|\mathbf{y})}\right]^w.$$

Here we explicitly note that the distributions are associated with the random variable $X_t$ in $R_t(\mathbf{x}_t, \mathbf{y})$. It is important to note that $R_t(\mathbf{x}_t, \mathbf{y})$ defined in Eq. (9) with $t = 0$ recovers the previously defined in Eq. (6), and that $p_{\phi, X_1}(\mathbf{y}|X_1 = \mathbf{a})$ and $p_{\phi, X_2}(\mathbf{y}|X_2 = \mathbf{a})$ are not necessarily identical, even for the same $(\mathbf{a}, \mathbf{y})$ value. To simplify notation, we will omit $X_t$ in subsequent expressions, as it should be clear from the context. Taking the derivative of the logarithm of Eq. (8) and applying the score function formula in Eq. (7) yields:

$$\bar{\boldsymbol{\epsilon}}_{\theta,t} = \boldsymbol{\epsilon}_{\theta,t} - \sqrt{1 - \bar{\alpha}_t}\nabla_{\mathbf{x}_t} \log R_t(\mathbf{x}_t, \mathbf{y}). \quad (10)$$

Substituting the rewards from Eq. (9) into Eq. (10) leads to Eq. (4). In other words, the reformulated goal of scaling all marginal distributions in Eq. (8) aligns with the guidance practice described in Eq. (4).

---

[1]CG and CFG rewards are explicitly stated in (Ho & Salimans, 2022), while AutoG reward is unspecified in (Karras et al., 2024a) and we reformulate it within this framework. Another subtly is that strictly speaking, the original goal of CG and CFG stated in (Ho & Salimans, 2022) correspond to Eqs. (8) and (9) in our paper (i.e., scaling all marginal distributions). We slightly abuse the concept and start from scaling only the terminal marginal Eq. (5) for presentation clarity.

[2]To align with the fact that Eq. (4) is also applied at $t = T$, here we must introduce $R_T$ and enforce scaling at $t = T$. However, this suggests the denoising starts from a distribution other than the fixed standard Gaussian. See § 4 for more discussion.

However, this reformulated goal itself is invalid, because we observe that once $R_t$ is given, $R_{t-1}$ is implicitly defined up to a normalization constant due to the denoising process:

$$R_{t-1}(\mathbf{x}_{t-1}, \mathbf{y}) \propto \frac{\mathbb{E}\left[\mathcal{N}(\mathbf{x}_{t-1}|\bar{\boldsymbol{\mu}}_{\theta,t}, \sigma_t^2 \mathbf{I}) R_t(\mathbf{x}_t, \mathbf{y})\right]}{\mathbb{E}\left[\mathcal{N}(\mathbf{x}_{t-1}|\boldsymbol{\mu}_{\theta,t}, \sigma_t^2 \mathbf{I})\right]} \quad (11)$$

Unless explicitly stated otherwise, all expectations in this paper are taken with respect to $p_\theta(\mathbf{x}_t|\mathbf{y})$. The expression of $\boldsymbol{\mu}_{\theta,t}$ is provided in Eq. (3), and $\bar{\boldsymbol{\mu}}_{\theta,t}$ is defined as follows:

$$\bar{\boldsymbol{\mu}}_{\theta,t} = \boldsymbol{\mu}_{\theta,t} + \frac{1 - \alpha_t}{\sqrt{\alpha_t}} \nabla_{\mathbf{x}_t} R_t(\mathbf{x}_t, \mathbf{y}). \quad (12)$$

See Appendix B for the proof. This observation leads to two significant corollaries. Firstly, the scaled objective presented in Eq. (8) is invalid as it imposes excessive constraints that may not be satisfied. Secondly, the original formulation in Eq. (5) is also invalid. Because it implicitly assumes that all $R_t$ (where $t = T, T-1, \cdots, 1$) should be identity functions, which consequently makes $R_0$ an identity function as well. Essentially, these two corollaries state that it is not feasible to construct a new DDPM corresponding to either the scaled objective in Eq. (5) or Eq. (8).

Finally, if we preserve $\bar{p}_\theta(\mathbf{x}_T|\mathbf{y})$ unchanged from $p_\theta(\mathbf{x}_T|\mathbf{y})$, i.e., $R_T$ is an identity function, then all subsequent rewards $R_t$ (where $t = T-1, T-2, \cdots, 1$) will also be identity functions based on Eq. (11). This implies that strictly adhering to the guidance theory requires a modification of the distribution at the denoising start. However, the chain start is typically fixed to a standard Gaussian distribution, irrespective of whether guidance is applied. We will revisit and elaborate on this subtlety in § 4 after reconstructing the guidance theory on a correct foundation.

## 4. Guidance Theory from Joint Scaling

**Scaling the Joint Distribution.** Examining the interpretation of guidance in the previous section, we identify the critical flaw lies in that it scales the *marginal* distributions, whereas the *joint* distribution should serve as the cornerstone of our analysis. Specifically, to build the correct guidance theory, we begin with a joint distribution scaled objective:

$$\bar{p}_\theta(\mathbf{x}_{0:T}|\mathbf{y}) \propto p_\theta(\mathbf{x}_{0:T}|\mathbf{y}) \cdot R_0(\mathbf{x}_0, \mathbf{y}). \quad (13)$$

Eq. (13) is similar to Eq. (5) in that the reward value depends only on the final generated sample $\mathbf{x}_0$ and the conditioning variable $\mathbf{y}$. However, it differs from Eq. (5) in that the reward influences the entire denoising chain $\mathbf{x}_{0:T}$. Furthermore, Eq. (13) can derive Eq. (5) by marginalizing out $\mathbf{x}_{1:T}$. Consequently, both of them have the same impact to the generation, since the generation is determined solely by the marginal $\bar{p}_\theta(\mathbf{x}_0|\mathbf{y})$. Before moving forward, we define the

induced expected reward $E_t(\mathbf{x}_t, \mathbf{y})$ at time step $t$ as:

$$E_t(\mathbf{x}_t, \mathbf{y}) = \int p_\theta(\mathbf{x}_0|\mathbf{x}_t, \mathbf{y}) R_0(\mathbf{x}_0, \mathbf{y}) \, d\mathbf{x}_0, \quad (14)$$

where $t = 0, 1, \cdots, T$. Note that the definition gives $E_0(\mathbf{x}_0, \mathbf{y}) = R_0(\mathbf{x}_0, \mathbf{y})$ at $t = 0$. For later simplicity, we introduce $\mathbf{x}_{T+1} = \varnothing$, and thus $E_{T+1}(\mathbf{x}_{T+1}, \mathbf{y}) = \int p_\theta(\mathbf{x}_0|\mathbf{y}) R_0(\mathbf{x}_0, \mathbf{y}) \, d\mathbf{x}_0$ based on Eq. (14). Since this expression does not depend on any specific $\mathbf{x}_t$ or $t$, we also denote it as $E_{T+1}(\mathbf{x}_{T+1}, \mathbf{y}) = E(\mathbf{y})$ and will use them interchangeably in subsequent discussions.

We now summarize our main result in Theorem 4.1, with the proof deferred to Appendix C. Theorem 4.1 introduces a scaled joint distribution $\bar{p}_\theta(\mathbf{x}_{0:T}|\mathbf{y})$ that we aim to construct, justifying the existence and uniqueness of the transition kernels corresponding to it, and showing the form of the marginals $\bar{p}_\theta(\mathbf{x}_t|\mathbf{y})$ and the updated noise prediction network $\bar{\boldsymbol{\epsilon}}^\star_{\theta,t}$. It implies that unlike the marginal scaling shown in Eqs. (5) and (8), the joint scaled goal $\bar{p}_\theta(\mathbf{x}_{0:T}|\mathbf{y})$ is valid since a new DDPM can be constructed to realize it.

**Theorem 4.1.** *To satisfy the scaled goal given in Eq. (13), we must have an unique set of transition kernels:*

$$\bar{p}_\theta(\mathbf{x}_t|\mathbf{x}_{t+1}, \mathbf{y}) = \frac{E_t(\mathbf{x}_t, \mathbf{y})}{E_{t+1}(\mathbf{x}_{t+1}, \mathbf{y})} p_\theta(\mathbf{x}_t|\mathbf{x}_{t+1}, \mathbf{y}), \quad (15)$$

*where $t = 0, 1, \cdots, T$ and $\mathbf{x}_T = \varnothing$, which also determines:*

$$\bar{p}_\theta(\mathbf{x}_t|\mathbf{y}) = \frac{E_t(\mathbf{x}_t, \mathbf{y})}{E(\mathbf{y})} p_\theta(\mathbf{x}_t|\mathbf{y}). \quad (16)$$

*It implies the noise prediction network should be:*

$$\bar{\boldsymbol{\epsilon}}^\star_{\theta,t} = \boldsymbol{\epsilon}_{\theta,t} - \sqrt{1 - \bar{\alpha}_t} \nabla_{\mathbf{x}_t} \log E_t(\mathbf{x}_t, \mathbf{y}). \quad (17)$$

**Important Remarks.** Surprisingly, Theorem 4.1 states that strictly adhering to the guidance theory must modify the denoising start distribution from $p_\theta(\mathbf{x}_T|\mathbf{y}) = \mathcal{N}(\mathbf{x}_T|\mathbf{0}, \mathbf{I})$ to $\bar{p}_\theta(\mathbf{x}_T|\mathbf{y}) = E_T(\mathbf{x}_T, \mathbf{y})/E(\mathbf{y}) \cdot \mathcal{N}(\mathbf{x}_T|\mathbf{0}, \mathbf{I})$. [3] Whether this adjustment should be adopted in practice remains an open question, as the new distribution $\bar{p}_\theta(\mathbf{x}_T|\mathbf{y})$ is generally unknown and analytically intractable for sampling. We leave this question for future work and conclude this topic by discussing a notable special case. Specifically, when $E_T(\mathbf{x}_T, \mathbf{y})$ varies slowly with respect to $\mathbf{x}_T$, the ratio $E_T(\mathbf{x}_T, \mathbf{y})/E(\mathbf{y}) \approx 1$ since $E(\mathbf{y}) = \mathbb{E}_{\mathbf{x}_T \sim \mathcal{N}(\mathbf{x}_T|\mathbf{0}, \mathbf{I})}[E_T(\mathbf{x}_T, \mathbf{y})]$. In this case, $\bar{p}_\theta(\mathbf{x}_T|\mathbf{y})$ can still be well approximated as a standard Gaussian.

Although Eq. (17) elegantly defines the updated noise prediction $\bar{\boldsymbol{\epsilon}}^\star_{\theta,t}$, it is infeasible to calculate. This is because the

---

[3]To our knowledge, we are the first to theoretically confirm that guidance impacts the denoising start distribution. A prior study (Wallace et al., 2023), sharing our spirit, attempt to optimize the noise distribution as a form of guidance.

term $E_t(\mathbf{x}_t, \mathbf{y})$ at time $t$ requires foresight into the future — it necessitates the complete execution of the denoising process to the terminal point $t = 0$. Alternatively, replacing $\nabla_{\mathbf{x}_t} \log E_t(\mathbf{x}_t, \mathbf{y})$ with $\nabla_{\mathbf{x}_t} \log R_t(\mathbf{x}_t, \mathbf{y})$ in Eq. (17) yields Eq. (10), which is precisely what current guidance practices employ. Hence, we posit that the original guidance practice is **best interpreted as an approximation** of the optimal noise prediction network, operating under joint scaling with the constraint of no foresight into the future. Importantly, our interpretation reconciles previous confusions (Chidambaram et al., 2024; Bradley & Nakkiran, 2024) regarding why guidance implementations do not result in sampling from $\bar{p}_\theta$, as Eq. (17) represents the optimal solution for sampling from $\bar{p}_\theta$, while the practical implementation is an approximation shown in Eq. (10).

The effectiveness of guidance hinges on well-designed rewards $R_t$, ensuring that $\bar{\epsilon}_{\theta,t} \approx \bar{\epsilon}_{\theta,t}^\star$ or equivalently $\nabla_{\mathbf{x}_t} \log R_t(\mathbf{x}_t, \mathbf{y}) \approx \nabla_{\mathbf{x}_t} \log E_t(\mathbf{x}_t, \mathbf{y})$. To quantify the approximation error, we present Theorem 4.2 and 4.3, with proofs deferred to Appendix D. Notably, these theorems and their proofs remain valid within the continuous-time SDE formulation of diffusion models. When $t$ is treated as continuous, Theorem 4.2 states that the mean squared approximation error decreases as the denoising process progresses ($t \to 0$ and $\Delta \to 0$), as outlined in Eq. (18). Moreover, the expected error is approximately linked to the second moment based on Eq. (19). Additionally, Theorem 4.3 states that the approximation still exhibits a bias even under a looser expectation perspective.

**Theorem 4.2.** *Assume a deterministic sampler is used in the reverse denoising process, and the original noise network $\epsilon_{\theta,t}$ is $L$-Lipschitz continuous with values bounded by $B$. For CFG and AutoG , the following bound holds:*

$$\sqrt{1 - \bar{\alpha}_t} \|\nabla_{\mathbf{x}_t} \log E_t(\mathbf{x}_t, \mathbf{y}) - \nabla_{\mathbf{x}_t} \log R_t(\mathbf{x}_t, \mathbf{y})\|$$
$$\leq 2wB\|\frac{d\Delta}{d\mathbf{x}_t}\| + 2wL\|\Delta\| + 2wLt , \quad (18)$$

*where $\Delta = \hat{\mathbf{x}}_0 - \mathbf{x}_t$, and $\hat{\mathbf{x}}_0$ represents the estimate of $\mathbf{x}_0$ based on $\mathbf{x}_t$ at time step $t$. Moreover, under these assumptions, we approximately have:*

$$\mathbb{E}[\|\bar{\epsilon}_{\theta,t}^\star - \bar{\epsilon}_{\theta,t}\|] \lesssim C_1 \mathbb{E}[\|\mathbf{x}_t\|] + C_2 , \quad (19)$$

*where $C_1$ and $C_2$ are constants of $\mathbf{x}_t$.*

**Theorem 4.3.** *The bias of approximation is:*

$$\mathbb{E}[\bar{\epsilon}_{\theta,t}^\star - \bar{\epsilon}_{\theta,t}] = \mathbb{E}[\epsilon_{\theta,t} \cdot \log \frac{R_t(\mathbf{x}_t, \mathbf{y})}{E_t(\mathbf{x}_t, \mathbf{y})}] . \quad (20)$$

**REG as an Alternative.** Upon recognizing that the original guidance technique in Eq. (10) is an approximation to the true guidance equation in Eq. (17), we attempt to explore whether alternative approximations exist. To this end, we propose a novel rectification to Eq. (10) by incorporating a gradient term, dubbed rectified gradient guidance (REG):

$$\bar{\epsilon}_{\theta,t}^{\text{REG}} = \epsilon_{\theta,t} - \sqrt{1 - \bar{\alpha}_t} \nabla_{\mathbf{x}_t} \log R_t(\mathbf{x}_t, \mathbf{y})$$
$$\odot \underbrace{\left(1 - \sqrt{1 - \bar{\alpha}_t} \frac{\partial(\mathbf{1}^T \cdot \epsilon_{\theta,t})}{\partial \mathbf{x}_t}\right)}_{\text{REG correction term}} , \quad (21)$$

where $\odot$ represents the element-wise product, and $\mathbf{1}^T \cdot \epsilon_{\theta,t}$ computes the sum of all elements in $\epsilon_{\theta,t}$. To motivate our above guidance equation, we first notice that $E_t(\mathbf{x}_t, \mathbf{y})$ in Eq. (17) becomes $R_0(\hat{\mathbf{x}}_0, \mathbf{y})$ when a deterministic sampler is adopted in the denoising process. If we further apply the chain rule, we obtain:

$$\bar{\epsilon}_{\theta,t}^\star = \epsilon_{\theta,t} - \sqrt{1 - \bar{\alpha}_t} \nabla_{\mathbf{x}_t} \log R_0(\hat{\mathbf{x}}_0, \mathbf{y})$$
$$= \epsilon_{\theta,t} - \sqrt{1 - \bar{\alpha}_t} \cdot \frac{\partial \hat{\mathbf{x}}_0}{\partial \mathbf{x}_t} \cdot \nabla_{\hat{\mathbf{x}}_0} \log R_0(\hat{\mathbf{x}}_0, \mathbf{y}) . \quad (22)$$

We have performed several approximations to get Eq. (21) from Eq. (22). Firstly, we replace $\nabla_{\mathbf{x}_0} \log R_0(\hat{\mathbf{x}}_0, \mathbf{y})$ with $\nabla_{\mathbf{x}_t} \log R_t(\hat{\mathbf{x}}_t, \mathbf{y})$. It is straightforward to show that their mean squared error is upper bounded by $2wL\|\Delta\| + 2wLt$ following the proof for Theorem 4.2 in Appendix D. Notably, this bound has one fewer term than the original guidance approximation error shown in Eq. (18). Secondly, we have used the analytical expression $\hat{\mathbf{x}}_0 = \frac{1}{\sqrt{\bar{\alpha}_t}}(\mathbf{x}_t - \sqrt{1 - \bar{\alpha}_t} \epsilon_{\theta,t})$ given by the DDPM formulation to simplify the jacobian matrix $\partial \hat{\mathbf{x}}_0 / \partial \mathbf{x}_t$, during which the ratio $1/\sqrt{\bar{\alpha}_t}$ has been omitted since it can be absorbed into the guidance strength $w$. It is important to note that different diffusion formulations (Karras et al., 2022) may produce slightly different analytical expressions relating $\mathbf{x}_t$ to $\hat{\mathbf{x}}_0$, leading to variations in how the Jacobian matrix $\partial \hat{\mathbf{x}}_0 / \partial \mathbf{x}_t$ is simplified compared to the DDPM case, which will be covered in our experiments in § 5.2. Finally, the last line of Eq. (22) involves a $D$-by-$D$ Jacobian matrix $\partial \hat{\mathbf{x}}_0 / \partial \mathbf{x}_t$ multiplied by a $D$-dimensional vector $\nabla_{\hat{\mathbf{x}}_0} \log R_0(\hat{\mathbf{x}}_0, \mathbf{y})$, which is computational prohibitive. We have simplified this operation by approximating it with element-wise vector multiplication in Eq. (21), which performs well empirically. This approximation can alternatively be interpreted as assuming that the Jacobian matrix is (approximately) diagonal.

## 5. Experimental Results

### 5.1. 1D and 2D Synthetic Examples

In this section, we conduct qualitative experiments on 1D and 2D synthetic examples. Based on the definition in Eq. (14), the exact calculation of $\nabla_{\mathbf{x}_t} \log E_t(\mathbf{x}, \mathbf{y})$ involves gradient propagation through the denoising chain, which is generally computationally intensive but can be affordably computed in 1D or 2D.

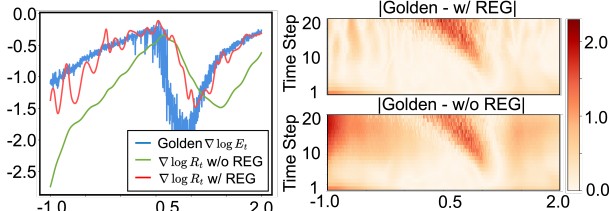

*Figure 1.* Left: Guidance values are plotted along the X-axis in the range $[-1.0, 2.0]$ at time step $t = 13$. Right: Heatmaps depict the absolute differences between each gradient guidance value and the optimal guidance $\nabla \log E_t$, plotted on uniform grids in $[-1.0, 2.0]$ at each time step. These two figures justify that our proposed REG aligns better with the optimal guidance $\nabla \log E_t$ compared to the vanilla CFG, i.e., $\nabla \log R_t$ without REG.

In the 1D example inspired by (Kynkäänniemi et al., 2024), we attempt to learn a conditional data distribution $q(x|y = 0) = 0.5 \times \mathcal{N}(x|0.5, 0.25^2) + 0.5 \times \mathcal{N}(x|1.5, 0.25^2)$ using a diffusion model with 20 steps and an MLP as the noise prediction network. The unconditional data distribution $q(x)$ is set as $0.5 \times \mathcal{N}(x| - 1, 0.5^2) + 0.5 \times \mathcal{N}(x|1, 0.5^2)$. As in (Kynkäänniemi et al., 2024), the number of classes and other class-conditioned distributions are considered irrelevant in this context. The reward $R_0(x, y) = p_\theta(x|y)/p_\theta(x)$ is chosen in Eq. (13). Following Eq. (14), the optimal $\nabla \log E_t$ [4] is the derivative of a scalar to a scalar, and can be computed via numerical integration at each time step $t$. Furthermore, we evaluate the original $\nabla \log R_t$ as per Eq. (10) and the $\nabla \log R_t$ corrected with REG as per Eq. (21). The results are visualized in Figure 1. On the right, $\nabla \log R_t$ with REG aligns more closely with the optimal solution $\nabla \log E_t$ across all time steps and grid points, compared to the version without REG. The left panel of Figure 1 highlights a specific case at $t = 13$. Experimental setup details and additional results are provided in Appendix E.1.

In the 2D example, we design a two-class conditional image generation task inspired by (Pärnamaa, 2023). The training dataset consists of 8,000 samples per class, represented as pairs $(\mathbf{x}_0, y)$, where $\mathbf{x}_0$ is a point in a 2D plane, and $y$ is either 0 or 1. We train a diffusion model with 25 time steps using a simple MLP as the noise prediction network. In this example, both $\nabla \log E_t$ and $\nabla \log R_t$ represent the derivative of a scalar with respect to a 2D vector and can be visualized as gradient arrows in a 2D plane. Figure 2 presents the results. Columns (a)-(c) in Figure 2 demonstrate that CFG with REG produces better results compared to without REG. Additionally, for both classes, column (e) is lighter in color and features coarser arrows compared to column (f), making it visually closer to the optimal column (d). The red-highlighted region for the first class provides

---

[4]For brevity and where no confusion arises, we will refer to $\nabla_{\mathbf{x}_t} \log E_t(\mathbf{x}_t, \mathbf{y})$ as $\nabla \log E_t$, and similarly for $\nabla \log R_t$.

further evidence to support this observation. To rigorously substantiate our claim, we present the win ratios in Table 1, indicating the cases where the error with REG and without REG is smaller, respectively. Finally, Figure 2 (g)-(h) display the magnitude of our REG correction term. We observe that the REG correction term exhibits rough vertical and horizontal patterns for the X-axis and Y-axis, respectively, as well as a "low-resolution" shape associated with each class. This behavior is expected because, when away from the target shape, the REG X-correction term, which relates to a derivative with respect to the X-coordinate, is only weakly dependent on the Y-coordinate. Consequently, it exhibits a vertical pattern, as shown in column (g). In contrast, near the target shape, data points occur at varying Y-coordinates for a fixed X-coordinate, causing the REG X-correction term to vary accordingly.

*Table 1.* The win ratio $x\% : y\%$ is reported for different time steps and classes, where $x\%$ and $y\%$ denote the cases where $\nabla \log R_t$ with REG and without REG achieve a smaller error, respectively.

| Time Step | Class 1 | Class 2 |
|---|---|---|
| $t = 20$ | 60.2% : 39.8% | 65.2% : 34.8% |
| $t = 16$ | 56.7% : 43.3% | 68.2% : 31.8% |
| $t = 12$ | 67.8% : 32.2% | 74.4% : 25.6% |
| $t = 8$ | 67.7% : 32.3% | 73.1% : 26.9% |
| $t = 4$ | 65.5% : 34.5% | 72.0% : 28.0% |

### 5.2. Quantitative Results: Image Generation

In this section, we perform quantitative experiments on class-conditional ImageNet generation and text-to-image generation. We consider state-of-the-art guidance techniques, including vanilla CFG (Ho & Salimans, 2022), cosine CFG (Gao et al., 2023), linear CFG, interval CFG (Kynkäänniemi et al., 2024), AutoG (Karras et al., 2024a), and demonstrate that our REG is a method-agnostic approach capable of enhancing the performance of all these techniques when used in conjunction with them. As summarized in Table 2, we deliberately select models spanning diverse architectures, various samplers (e.g., DDPM, Euler, and 2nd Heun), and different prediction parameterizations (e.g., noise prediction and $\mathbf{x}_0$ prediction), to rigorously verify the effectiveness of our proposed REG method.

**Class-Conditional ImageNet Generation.** We evaluate various resolutions, including $64 \times 64$, $256 \times 256$, and $512 \times 512$, using DiT (Peebles & Xie, 2023) and EDM2 (Karras et al., 2024b) as baseline models. Since EDM2 employs a different analytical relationship between $\mathbf{x}_t$ and $\mathbf{x}_0$ compared to the DDPM formulation, the REG correction term must be re-derived. See Appendix E.2 for details. Fréchet Inception Distance (FID) and Inception score (IS) are the

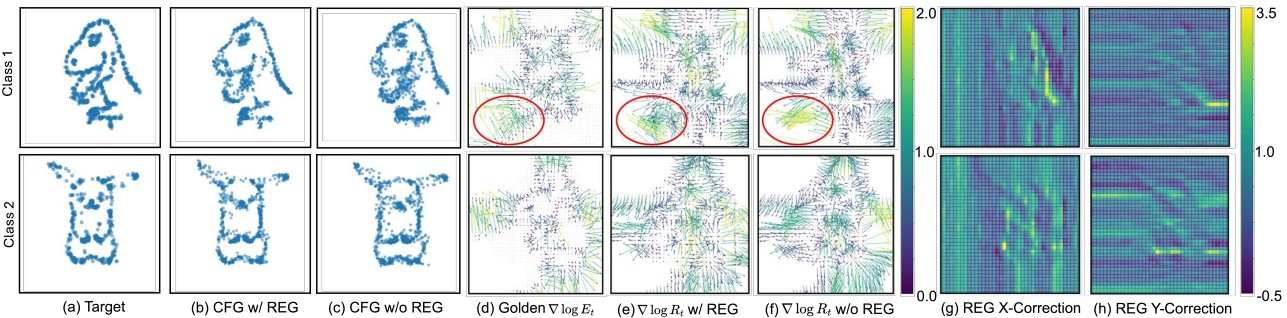

*Figure 2.* Results of guidance on a synthetic 2D two-class conditional generation task using a simple diffusion model with 25 time steps. (a)-(c) illustrate the target shape to be learned, the shape generated using CFG with our proposed REG, and the shape generated without REG, respectively. (d)-(f) depict $\nabla \log E_t$ and $\nabla \log R_t$ at $t = 9$, which are gradients of a scalar with respect to a 2D vector, visualized as arrows in a 2D plane. (g)-(h) show the magnitude of the REG correction term (i.e., the second line of Eq. (21)) at $t = 9$.

*Table 2.* A summary of the models used in our experiment.

| Model | # Param | Sampler | Prediction |
|---|---|---|---|
| DiT-XL/2 | 675 M | 250-step DDPM | $\epsilon$-prediction |
| EDM2-S | 280 M | 2nd Heun | $\mathbf{x}_0$-prediction |
| EDM2-XXL | 1.5 B | 2nd Heun | $\mathbf{x}_0$-prediction |
| SD-v1-4 | 860 M | PNDM | $\epsilon$-prediction |
| SD-XL | 2.6 B | Euler Discrete | $\epsilon$-prediction |

two key metrics used to evaluate the generated image quality and diversity, respectively. We sweep the guidance scale across a broad range to ensure coverage of the turning points on the FID and IS curves. The best FID achieved by each method is reported in Table 3 along with the corresponding IS. As shown in Table 3, cosine and linear CFG perform marginally better than vanilla CFG. On the other hand, interval CFG significantly outperforms vanilla CFG, but we also note that determining the optimal interval range involves a fine-grained and computationally expensive search.

Table 3 shows that incorporating our proposed REG, the FID usually decreases and IS increases across different models, resolutions, and guidance methods. Note that for EDM2-S and EDM2-XXL, interval CFG (Kynkäänniemi et al., 2024) applies guidance to around 20% of all the time steps, so the impact of REG is less pronounced in this case. Alternatively, when applying interval CFG to DiT (Kynkäänniemi et al., 2024), around 30% time steps use guidance, and REG shows meaningful improvement in this case reducing FID from 1.95 to 1.86. To better justify REG's efficacy, we plot the Pareto front of FID versus IS in Figure 3. It illustrates that using REG universally pushes the Pareto front downward and to the right, achieving more optimal FID and IS metrics. See Appendix E.2 for experimental details and extra results.

**Text-to-Image Generation.** We use pretrained stable diffusion model SD-v1-4 (Rombach et al., 2022), and SD-XL (Podell et al., 2023) available on Huggingface for this

*Table 3.* Class-conditional ImageNet generation results are reported. The red ↓ and green ↑ arrows indicate better FID and IS metrics, respectively, achieved by using REG with a specific established guidance method compared to not using REG.

| Resolution | Benchmark | FID ↓ | IS ↑ |
|---|---|---|---|
| 64×64 | **EDM2-S** | 1.580 | —— |
| | + AutoG | 1.044 | 69.01 |
| | + REG (ours) | 1.035 ↓ | 69.01 |
| 256×256 | **DiT-XL/2** | 9.62 | 121.50 |
| | + Vanilla CFG | 2.21 | 248.36 |
| | + REG (ours) | 2.04 ↓ | 276.26 ↑ |
| | + Cosine CFG | 2.30 | 300.73 |
| | + REG (ours) | 1.76 ↓ | 287.48 |
| | + Linear CFG | 2.23 | 268.69 |
| | + REG (ours) | 2.18 ↓ | 284.20 ↑ |
| | + Interval CFG | 1.95 | 250.44 |
| | + REG (ours) | 1.86 ↓ | 259.57 ↑ |
| 512×512 | **EDM2-S** | 2.56 | —— |
| | + Vanilla CFG | 2.29 | 268.56 |
| | + REG (ours) | 2.02 ↓ | 275.30 ↑ |
| | + Cosine CFG | 2.16 | 282.46 |
| | + REG (ours) | 1.99 ↓ | 291.77 ↑ |
| | + Linear CFG | 2.21 | 282.89 |
| | + REG (ours) | 1.99 ↓ | 291.04 ↑ |
| | + Interval CFG | 1.67 | 287.45 |
| | + REG (ours) | 1.67 | 288.43 ↑ |
| | **EDM2-XXL** | 1.91 | —— |
| | + Vanilla CFG | 1.83 | 265.76 |
| | + REG (ours) | 1.74 ↓ | 289.24 ↑ |
| | + Cosine CFG | 1.80 | 261.94 |
| | + REG (ours) | 1.69 ↓ | 268.84 ↑ |
| | + Linear CFG | 1.81 | 262.03 |
| | + REG (ours) | 1.69 ↓ | 268.30 ↑ |
| | + Interval CFG | 1.45 | 283.26 |
| | + REG (ours) | 1.45 | 288.72 ↑ |

experiment. Due to the exhaustive search required for the interval range in interval CFG and the need for a carefully constructed "bad" version in AutoG, these two methods are not practical within our computational budget and have been

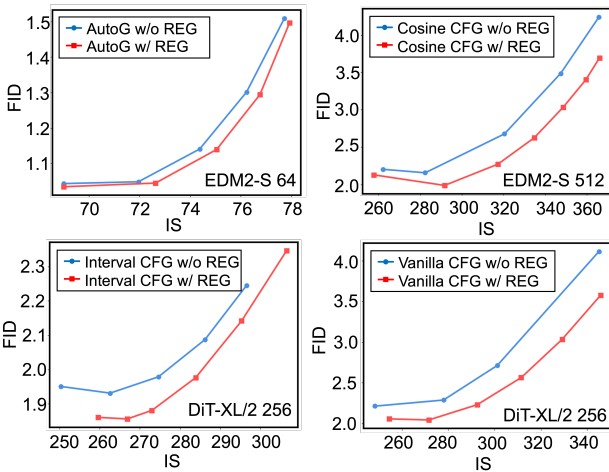

*Figure 3.* The Pareto front of FID versus IS is presented by varying the guidance weight $w$ over a broad range for different methods. The turning points are also included. Curves positioned further toward the bottom-right indicate superior performance. See Appendix E.2 for extra Pareto front results.

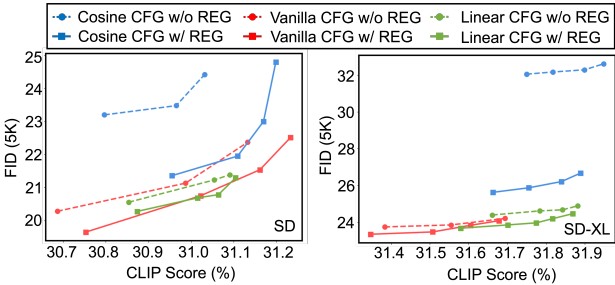

*Figure 4.* The Pareto front of FID versus CLIP score is shown by varying the guidance weight $w$ across a broad range for different methods. Curves closer to the bottom-right are better.

*Table 4.* Text-to-Image generation results on the COCO-2017 5K dataset are reported. The red ↓ and green ↑ arrows indicate better FID and CLIP score, respectively, achieved by using REG with a specific established guidance method compared to not using REG.

| Model | Benchmark | FID ↓ | CLIP (%) ↑ |
|---|---|---|---|
| SD-v1-4 512×512 | + Vanilla CFG | 20.27 | 30.68 |
| | + REG (ours) | 19.63 ↓ | 30.75 ↑ |
| | + Cosine CFG | 23.19 | 30.80 |
| | + REG (ours) | 21.35 ↓ | 30.96 ↑ |
| | + Linear CFG | 20.55 | 30.85 |
| | + REG (ours) | 20.27 ↓ | 30.87 ↑ |
| SD-XL 1024×1024 | + Vanilla CFG | 23.73 | 31.38 |
| | + REG (ours) | 23.46 ↓ | 31.51 ↑ |
| | + Cosine CFG | 32.14 | 31.58 |
| | + REG (ours) | 25.62 ↓ | 31.66 ↑ |
| | + Linear CFG | 24.43 | 31.55 |
| | + REG (ours) | 23.67 ↓ | 31.58 ↑ |

omitted. We evaluate different CFG approaches with and without REG on the COCO-2017 dataset (with 5,000 gen-

erated images) using FID and CLIP score as the evaluation metrics. We sweep a broad range of guidance scales, similar to the class-conditional ImageNet generation experiment, and present the results in Table 4.

Table 4 indicates that despite SD-XL having a larger parameter count, it appears to perform worse than SD-v1-4 on this specific COCO benchmark, a phenomenon observed in previous literature (Wang et al., 2024a;b). We also observe that cosine and linear CFG underperform compared to vanilla CFG in this setting, consistent with findings reported in prior work (Wang et al., 2024b). Nevertheless, Table 4 demonstrates that our REG can improve the FID and CLIP scores for all settings. Moreover, the Pareto fronts of FID versus CLIP score are shown in Figure 4, further justifying the effectiveness of REG. Qualitative results of generated images under different text prompts are deferred to Appendix E.2.

**Runtime and Memory Overhead.** Since REG requires the computation of an additional Jacobian matrix compared to the standard CFG approach, it naturally incurs higher memory usage and runtime. We also note that similar inference-time gradient calculations have also been explored in universal guidance (Bansal et al., 2024), albeit in a different context. Table 5 presents a summary of the runtime and memory overhead introduced by REG on a single NVIDIA A40 GPU under identical experimental settings, compared to vanilla CFG approach. In almost all settings, we have affordable less than 2× runtime and memory overhead.

*Table 5.* Comparison of CFG and REG runtime performance and peak GPU memory usage across different models, resolutions, and batch sizes. Memory is reported with batch size 1 to isolate per-image cost.

| Model | Resolution / BS | CFG / REG Runtime (sec) |
|---|---|---|
| EDM2-S | 64 / 8 | 25.96 / 42.99 (1.66×) |
| DiT-XL/2 | 256 / 8 | 59.79 / 94.23 (1.58×) |
| EDM2-S | 512 / 8 | 46.14 / 62.87 (1.36×) |
| EDM2-XXL | 512 / 8 | 49.21 / 92.60 (1.88×) |
| SD-v1-4 | 512 / 4 | 32.63 / 39.54 (1.21×) |
| SD-XL | 1024 / 2 | 47.48 / 74.52 (1.57×) |

| Model | Resolution / BS | CFG / REG Memory (GB) |
|---|---|---|
| EDM2-S | 64 / 1 | 0.87 / 1.49 (1.71×) |
| DiT-XL/2 | 256 / 1 | 4.15 / 5.01 (1.21×) |
| EDM2-S | 512 / 1 | 1.19 / 1.81 (1.52×) |
| EDM2-XXL | 512 / 1 | 4.59 / 7.31 (1.59×) |
| SD-v1-4 | 512 / 1 | 2.73 / 4.39 (1.61×) |
| SD-XL | 1024 / 1 | 6.91 / 19.49 (2.82×) |

## 6. Related Work

Guidance techniques are indispensable in modern diffusion models, significantly improving conditional generation quality. Classifier guidance (CG), introduced by Dhariwal &

Nichol (2021), improves the quality of conditional samples by using the gradient of an auxiliary classifier as a guidance signal, enabling a trade-off between the FID and IS metric. Classifier-free guidance (CFG) removes the need for an external classifier by training a noise prediction network to handle both conditional and unconditional generation via random conditioning dropout (Ho & Salimans, 2022). CFG has shown significant empirical success and inspired numerous extensions (Kynkäänniemi et al., 2024; Karras et al., 2024a; Bansal et al., 2024; Zheng & Lan, 2024; Chung et al., 2024; Ahn et al., 2024). Building on CFG, auto-guidance (AutoG) employs a "bad version" of the noise prediction network, such as a checkpoint from a lighter model, to produce the guidance signal (Karras et al., 2024a). AutoG achieves state-of-the-art FID and IS scores compared to other guidance methods. However, identifying an effective "bad version" is non-trivial and may require an exhaustive search. Universal guidance (Bansal et al., 2024) extends the framework by utilizing arbitrary guidance signals without retraining. While our work is in a completely different context — focusing on building a theoretically sound guidance theory — REG is similar to universal guidance in involving $\partial \epsilon_{\theta,t}/\partial \mathbf{x}_t$ in the implementation.

Theoretical analyses of unconditional and conditional diffusion models are abundant, yet the motivation behind guidance has been addressed only superficially in the literature (Dhariwal & Nichol, 2021; Ho & Salimans, 2022), with its theory primarily developed through Gaussian mixture case studies (Wu et al., 2024; Bradley & Nakkiran, 2024; Chidambaram et al., 2024). Guidance was commonly described as being motivated by sampling from a scaled distribution $\bar{p}_\theta$ that places greater emphasis on regions aligning with the conditioning variable. Leveraging the score function formula, it transforms sampling from the scaled distribution into an updated noise prediction network $\bar{\epsilon}_{\theta,t}$, which can be seamlessly integrated into the reverse denoising process. However, recent works (Bradley & Nakkiran, 2024; Chidambaram et al., 2024; Xia et al., 2024) challenge this interpretation by showing that using the modified $\bar{\epsilon}_{\theta,t}$ in the reverse denoising process does not yield sampling from $\bar{p}_\theta$. Our work attempts to reconcile this theory and practice discrepancy by rebuilding the correct guidance theory from scaling the joint distribution. We complement the findings of Bradley & Nakkiran (2024); Chidambaram et al. (2024); Xia et al. (2024) by providing the correct form of $\bar{\epsilon}_{\theta,t}^\star$ to ensure sampling from $\bar{p}_\theta$.

Another related and concurrent work (Pavasovic et al., 2025) demonstrates that, although CFG may reduce sample diversity in low-dimensional settings, it proves beneficial in high-dimensional regimes due to a 'blessing of dimensionality'. Their analysis reveals two distinct phases: an early phase where CFG aids in accurate class selection, followed by a later phase in which its impact becomes negligible.

## 7. Conclusions and Limitations

In this paper, we rebuild the guidance theory on the correct theoretical foundation based on joint scaling and reconcile the practice and theory gap. We demonstrate that guidance methods are an approximation to the optimal solution under no future foresight constraint. Leveraging this framework, we introduce rectified gradient guidance (REG), a versatile enhancement that consistently improves the performance of existing guidance techniques. Comprehensive experiments validate the effectiveness of REG. Our work resolves long-standing misconceptions about guidance methods and paves the way for its future advancements.

Our proposed REG is grounded in theoretical insights and mainly serves as an experimental validation of our theory's correctness. Its practical use depends on application needs, as it improves conditional generation performance with a minor computational overhead due to gradient calculations.

### Acknowledgements

Zhengqi Gao leads the theoretical development, algorithm design, and overall project coordination. Kaiwen Zha contributes to the theoretical analysis and experimental setup. Tianyuan Zhang and Zihui Xue contribute to the experimental design and evaluation. Duane S. Boning supervises the project and provides guidance throughout. The authors gratefully acknowledge the constructive feedback provided by the anonymous reviewers, which significantly improves the quality of this paper.

### Impact Statement

Our research focuses on building the correct guidance theory for conditional generation in diffusion models, ultimately contributing to improved image generation quality. We acknowledge that large-scale diffusion models pose significant societal risks, such as generating fake images and videos, exacerbating misinformation, amplifying biases, and undermining trust in visual media. While our method enhances the generative capabilities of diffusion models, it may inadvertently amplify these associated risks. To mitigate these concerns, we emphasize the importance of responsible development and deployment of diffusion models under stringent oversight. This includes, but is not limited to, integrating bias-reduction strategies and implementing advanced mechanisms for detecting and preventing misuse.

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

## A. Remarks on Score Function Formula

We will illustrate the score function formula for the unconditional generation case, corresponding to the second line of Eq. (7), noting that the first line of Eq. (7) associated with the conditional generation is similar.

The continuous-time SDE formulation of diffusion models (Song et al., 2021; Lu et al., 2022) describes the forward diffusion process as a stochastic differential equation (SDE) and establishes the existence of a reverse-time SDE that is mathematically equivalent to the forward-time SDE. The only unknown term in the reverse-time SDE is the score function $\nabla_{\mathbf{x}_t} \log q_t(\mathbf{x}_t)$, which is learned by the neural network $\epsilon_\theta(\mathbf{x}_t, t)$ through proper parameterization involving noise schedules during training. Notably, the score matching training objective uniformly samples $t$ over the entire range $[0, T]$ with a weighting scheme. This design ensures that the score function formula Eq. (7) holds approximately across the entire range $[0, T]$ by definition.

In contrast, the discrete-time DDPM formulation can, strictly speaking, only derive that Eq. (7) holds for $t = 1, 2, \cdots, T$, but not for $t = 0$. A straightforward explanation is as follows: note that $\epsilon_\theta(\mathbf{a}, t = 1)$ is the noise prediction at the final time step, used to update from $\mathbf{x}_1 = \mathbf{a}$ to $\mathbf{x}_0$, and $\epsilon_\theta(\cdot, t = 0)$ is never utilized in the reverse denoising process and unused during training. Consider an exaggerated case where $\epsilon_\theta(\cdot, t = 0) = \infty$, while $\epsilon_\theta(\cdot, t)$ for $t = 1, 2, \cdots, T$ behaves correctly and enables perfect generation. In this case, the left-hand side of Eq. (7), $\nabla_{\mathbf{x}_0} \log p_\theta(\mathbf{x}_0)$ equals $\nabla_{\mathbf{x}_0} \log q(\mathbf{x}_0)$ and remains proper and finite, while the right-hand side becomes infinite, illustrating the inconsistency of Eq. (7) for $t = 0$.

However, in practice, the score function formula Eq. (7) can be considered valid at $t = 0$ in the DDPM formulation, provided sufficiently dense time grids (i.e., a large enough $T$) are used. This is because Eq. (7) holds at $t = 1$, and for dense time grids, both sides of Eq. (7) at $t = 1$ closely approximate their counterparts at $t = 0$. Consequently, Eq. (7) effectively holds at $t = 0$. This perspective is also the discretized view of the SDE formulation. Our aim here is to highlight the correct way to understand the score function formula.

## B. Invalid Marginal Scaling

We will prove that $R_{t-1}$ is implicitly determined by $R_t$, i.e., Eq. (11), in this section. First, we calculate the derivative of logarithm of Eq. (8), yielding:

$$\nabla_{\mathbf{x}_t} \log \bar{p}_\theta(\mathbf{x}_t|\mathbf{y}) = \nabla_{\mathbf{x}_t} \log p_\theta(\mathbf{x}_t|\mathbf{y}) + \nabla_{\mathbf{x}_t} \log R(\mathbf{x}_t, \mathbf{y}). \tag{23}$$

Substituting the score function relationship shown in Eq. (7) into it, we obtain:

$$\bar{\epsilon}_{\theta,t} = \epsilon_{\theta,t} - \sqrt{1 - \bar{\alpha}_t} \nabla_{\mathbf{x}_t} \log R_t(\mathbf{x}_t, \mathbf{y}), \tag{24}$$

where we use the short notation $\bar{\epsilon}_{\theta,t} = \bar{\epsilon}_{\theta,t}(\mathbf{x}_t, t, \mathbf{y})$ and $\epsilon_\theta = \epsilon_\theta(\mathbf{x}_t, t, \mathbf{y})$ as in the main text. The modified transition kernel $\bar{p}_\theta(\mathbf{x}_{t-1}|\mathbf{x}_t, \mathbf{y})$ corresponding to this modified $\bar{\epsilon}_{\theta,t}$ can be derived by analogy to Eq. (2) and Eq. (3):

$$\bar{p}_\theta(\mathbf{x}_{t-1}|\mathbf{x}_t, \mathbf{y}) = \mathcal{N}(\mathbf{x}_{t-1}|\bar{\boldsymbol{\mu}}_{\theta,t}, \sigma_t^2 \mathbf{I}),$$
$$\bar{\boldsymbol{\mu}}_{\theta,t} = \frac{1}{\sqrt{\alpha_t}}\left(\mathbf{x}_t - \frac{1 - \alpha_t}{\sqrt{1 - \bar{\alpha}_t}}\bar{\epsilon}_\theta\right) = \boldsymbol{\mu}_{\theta,t} + \frac{1 - \alpha_t}{\sqrt{\alpha_t}}\nabla_{\mathbf{x}_t}\log R_t(\mathbf{x}_t, \mathbf{y}), \tag{25}$$

where Eq. (24) has been used to simplify $\bar{\boldsymbol{\mu}}_{\theta,t}$. Now, based on the defintion of the reverse denoising process, we have:

$$\bar{p}_\theta(\mathbf{x}_{t-1}|\mathbf{y}) = \int \bar{p}_\theta(\mathbf{x}_{t-1}|\mathbf{x}_t, \mathbf{y})\,\bar{p}_\theta(\mathbf{x}_t|\mathbf{y})\,d\mathbf{x}_t,$$
$$\Rightarrow p_\theta(\mathbf{x}_{t-1}|\mathbf{y})\,R_{t-1}(\mathbf{x}_{t-1}, \mathbf{y}) \propto \int \mathcal{N}(\mathbf{x}_{t-1}|\bar{\boldsymbol{\mu}}_{\theta,t}, \sigma_t^2 \mathbf{I})\,p_\theta(\mathbf{x}_t|\mathbf{y})R_t(\mathbf{x}_t, \mathbf{y})\,d\mathbf{x}_t,$$
$$\Rightarrow R_{t-1}(\mathbf{x}_{t-1}, \mathbf{y}) \propto \frac{1}{p_\theta(\mathbf{x}_{t-1}|\mathbf{y})}\int \mathcal{N}(\mathbf{x}_{t-1}|\bar{\boldsymbol{\mu}}_{\theta,t}, \sigma_t^2 \mathbf{I})\,p_\theta(\mathbf{x}_t|\mathbf{y})R_t(\mathbf{x}_t, \mathbf{y})\,d\mathbf{x}_t, \tag{26}$$
$$\Rightarrow R_{t-1}(\mathbf{x}_{t-1}, \mathbf{y}) \propto \frac{\int \mathcal{N}(\mathbf{x}_{t-1}|\bar{\boldsymbol{\mu}}_{\theta,t}, \sigma_t^2 \mathbf{I})\,p_\theta(\mathbf{x}_t|\mathbf{y})\,R_t(\mathbf{x}_t, \mathbf{y})\,d\mathbf{x}_t}{\int \mathcal{N}(\mathbf{x}_{t-1}|\boldsymbol{\mu}_{\theta,t}, \sigma_t^2 \mathbf{I})\,p_\theta(\mathbf{x}_t|\mathbf{y})\,d\mathbf{x}_t},$$
$$\Rightarrow R_{t-1}(\mathbf{x}_{t-1}, \mathbf{y}) \propto \frac{\mathbb{E}_{p_\theta(\mathbf{x}_t|\mathbf{y})}\left[\mathcal{N}(\mathbf{x}_{t-1}|\bar{\boldsymbol{\mu}}_{\theta,t}, \sigma_t^2 \mathbf{I})R_t(\mathbf{x}_t, \mathbf{y})\right]}{\mathbb{E}_{p_\theta(\mathbf{x}_t|\mathbf{y})}\left[\mathcal{N}(\mathbf{x}_{t-1}|\boldsymbol{\mu}_{\theta,t}, \sigma_t^2 \mathbf{I})\right]}.$$

Eq. (26) implies that when $R_t$ is given, $R_{t-1}$ is implicitly determined up to a normalization constant. Note here we assume the co-variance matrix is fixed to $\sigma_t^2 \mathbf{I}$. Nevertheless, the same derivation can be applied to learnable co-variance matrix.

## C. Scaling the Joint Distribution: Proof of Theorem 4.1

Based on the discrete-DDPM formulation, we know the joint distribution is construcuted as a Markov chain:

$$\bar{p}_\theta(\mathbf{x}_{0:T}|\mathbf{y}) = \bar{p}(\mathbf{x}_T|\mathbf{y}) \prod_{t=1}^{T} \bar{p}_\theta(\mathbf{x}_{t-1}|\mathbf{x}_t, \mathbf{y}). \tag{27}$$

When $\bar{p}_\theta(\mathbf{x}_{0:T}|\mathbf{y})$ is given, the decomposition to the initial $\bar{p}(\mathbf{x}_T|\mathbf{y})$ and the $T$ transitions $\bar{p}_\theta(\mathbf{x}_{t-1}|\mathbf{x}_t, \mathbf{y})$ is unique. This is because $\bar{p}_\theta(\mathbf{x}_{0:T}|\mathbf{y})$ is given, by doing marginalization, we obtain unique $\bar{p}_\theta(\mathbf{x}_{t-1}, \mathbf{x}_t|\mathbf{y})$ and $\bar{p}_\theta(\mathbf{x}_t|\mathbf{y})$. Then, based on the conditional probability definition: $\bar{p}_\theta(\mathbf{x}_{t-1}|\mathbf{x}_t, \mathbf{y}) = \bar{p}_\theta(\mathbf{x}_{t-1}, \mathbf{x}_t|\mathbf{y})/\bar{p}_\theta(\mathbf{x}_t|\mathbf{y})$, the transition kernel is also unique.

Thus, it suffices to demonstrate that the transition kernels presented in Eq. (15) satisfy the joint distribution specified in Eq. (13) and that each kernel represents a valid distribution. Once these conditions are established, Eq. (15) is guaranteed to be the unique solution. Now, we can derive:

$$\bar{p}_\theta(\mathbf{x}_{0:T}|\mathbf{y}) = \bar{p}_\theta(\mathbf{x}_T|\mathbf{y}) \prod_{t=0}^{T-1} \bar{p}_\theta(\mathbf{x}_t|\mathbf{x}_{t+1}, \mathbf{y}) = \prod_{t=0}^{T} \bar{p}_\theta(\mathbf{x}_t|\mathbf{x}_{t+1}, \mathbf{y}) = \prod_{t=0}^{T} \frac{E_t(\mathbf{x}_t, \mathbf{y})}{E_{t+1}(\mathbf{x}_{t+1}, \mathbf{y})} p_\theta(\mathbf{x}_t|\mathbf{x}_{t+1}, \mathbf{y})$$

$$= \frac{E_0(\mathbf{x}_0, \mathbf{y})}{E_{T+1}(\mathbf{x}_{T+1}, \mathbf{y})} \prod_{t=0}^{T} p_\theta(\mathbf{x}_t|\mathbf{x}_{t+1}, \mathbf{y}) = \frac{R_0(\mathbf{x}_0, \mathbf{y})}{E(\mathbf{y})} p_\theta(\mathbf{x}_{0:T}|\mathbf{y}) \propto p_\theta(\mathbf{x}_{0:T}|\mathbf{y}), \tag{28}$$

where we have used the notational convention $\mathbf{x}_{T+1} = \varnothing$ introduced earlier, $E_0(\mathbf{x}_0, \mathbf{y}) = R_0(\mathbf{x}_0, \mathbf{y})$, and $E_{T+1}(\mathbf{x}_{T+1}, \mathbf{y}) = E(\mathbf{y})$. The above equation states $\bar{p}_\theta(\mathbf{x}_{0:T}|\mathbf{y}) \propto R_0(\mathbf{x}_0, \mathbf{y})p_\theta(\mathbf{x}_{0:T}|\mathbf{y})$, with $1/E(\mathbf{y})$ being the normalization constant. Next, we proceed to demonstrate that Eq. (15) represents a valid distribution:

$$\int \bar{p}_\theta(\mathbf{x}_t|\mathbf{x}_{t+1}, \mathbf{y})d\mathbf{x}_t = \int \frac{E_t(\mathbf{x}_t, \mathbf{y})}{E_{t+1}(\mathbf{x}_{t+1}, \mathbf{y})} p_\theta(\mathbf{x}_t|\mathbf{x}_{t+1}, \mathbf{y})d\mathbf{x}_t$$

$$= \frac{1}{E_{t+1}(\mathbf{x}_{t+1}, \mathbf{y})} \int E_t(\mathbf{x}_t, \mathbf{y})p_\theta(\mathbf{x}_t|\mathbf{x}_{t+1}, \mathbf{y})d\mathbf{x}_t$$

$$= \frac{1}{E_{t+1}(\mathbf{x}_{t+1}, \mathbf{y})} \int p_\theta(\mathbf{x}_0|\mathbf{x}_t, \mathbf{y})R_0(\mathbf{x}_0, \mathbf{y})p_\theta(\mathbf{x}_t|\mathbf{x}_{t+1}, \mathbf{y})d\mathbf{x}_t\, d\mathbf{x}_0 \tag{29}$$

$$= \frac{1}{E_{t+1}(\mathbf{x}_{t+1}, \mathbf{y})} \int p_\theta(\mathbf{x}_0, \mathbf{x}_t|\mathbf{x}_{t+1}, \mathbf{y})R_0(\mathbf{x}_0, \mathbf{y})d\mathbf{x}_t\, d\mathbf{x}_0$$

$$= \frac{1}{E_{t+1}(\mathbf{x}_{t+1}, \mathbf{y})} \int p_\theta(\mathbf{x}_0|\mathbf{x}_{t+1}, \mathbf{y})R_0(\mathbf{x}_0, \mathbf{y})\, d\mathbf{x}_0 = \frac{E_{t+1}(\mathbf{x}_{t+1}, \mathbf{y})}{E_{t+1}(\mathbf{x}_{t+1}, \mathbf{y})} = 1.$$

Next, we use induction to prove Eq. (16). To begin with, substituting $t = T$ into Eq. (15) and noticing our introduced convention $\mathbf{x}_{T+1} = \varnothing$, we obtain $\bar{p}_\theta(\mathbf{x}_T|\mathbf{y}) = E_T(\mathbf{x}_T, \mathbf{y})/E(\mathbf{y}) \cdot p_\theta(\mathbf{x}_T|\mathbf{y})$, which aligns with Eq. (16) at $t = T$. Now suppose Eq. (16) holds for $t$, then for $t - 1$, we have:

$$\bar{p}_\theta(\mathbf{x}_{t-1}|\mathbf{y}) = \int \bar{p}_\theta(\mathbf{x}_{t-1}|\mathbf{x}_t, \mathbf{y})\bar{p}_\theta(\mathbf{x}_t|\mathbf{y})\, d\mathbf{x}_t = \int \frac{E_{t-1}(\mathbf{x}_{t-1}, \mathbf{y})}{E_t(\mathbf{x}_t, \mathbf{y})} p_\theta(\mathbf{x}_{t-1}|\mathbf{x}_t, \mathbf{y}) \frac{E_t(\mathbf{x}_t, \mathbf{y})}{E(\mathbf{y})} p_\theta(\mathbf{x}_t|\mathbf{y})\, d\mathbf{x}_t$$

$$= \int \frac{E_{t-1}(\mathbf{x}_{t-1}, \mathbf{y})}{E(\mathbf{y})} p_\theta(\mathbf{x}_{t-1}, \mathbf{x}_t|\mathbf{y})\, d\mathbf{x}_t = \frac{E_{t-1}(\mathbf{x}_{t-1}, \mathbf{y})}{E_t(\mathbf{y})} p_\theta(\mathbf{x}_{t-1}|\mathbf{y}), \tag{30}$$

which completes the induction. Finally, Eq. (17) is straightforward to prove by calculating the derivative of logarithm of Eq. (16) and using the score function relationship shown in Eq. (7):

$$\bar{p}_\theta(\mathbf{x}_t|\mathbf{y}) = \frac{E_t(\mathbf{x}_t, \mathbf{y})}{E(\mathbf{y})} p(\mathbf{x}_t|\mathbf{y})$$

$$\Rightarrow \log \bar{p}_\theta(\mathbf{x}_t|\mathbf{y}) = \log E_t(\mathbf{x}_t, \mathbf{y}) - \log E(\mathbf{y}) + \log p(\mathbf{x}_t|\mathbf{y})$$

$$\Rightarrow \nabla_{\mathbf{x}_t} \log \bar{p}_\theta(\mathbf{x}_t|\mathbf{y}) = \nabla_{\mathbf{x}_t} \log E_t(\mathbf{x}_t, \mathbf{y}) + \nabla_{\mathbf{x}_t} \log p(\mathbf{x}_t|\mathbf{y}) \tag{31}$$

$$\Rightarrow -\frac{\bar{\epsilon}_\theta^\star(\mathbf{x}_t, t, \mathbf{y})}{\sqrt{1 - \bar{\alpha}_t}} = \nabla_{\mathbf{x}_t} \log E_t(\mathbf{x}_t, \mathbf{y}) - \frac{\epsilon_\theta(\mathbf{x}_t, t, \mathbf{y})}{\sqrt{1 - \bar{\alpha}_t}}$$

$$\Rightarrow \bar{\epsilon}_\theta^\star(\mathbf{x}_t, t, \mathbf{y}) = \epsilon_\theta(\mathbf{x}_t, t, \mathbf{y}) - \sqrt{1 - \bar{\alpha}_t}\nabla_{\mathbf{x}_t} \log E_t(\mathbf{x}_t, \mathbf{y}).$$

## D. Approximation Error: Proof of Theorem 4.2 and Theorem 4.3

**Proof of Theorem 4.2.** We first notice that when using a deterministic sampler in the reverse denoising process, $E_t(\mathbf{x}_t, \mathbf{y})$ becomes $R_0(\hat{\mathbf{x}}_0, \mathbf{y})$, where $\hat{\mathbf{x}}_0$ is the estimate of $\mathbf{x}_0$ at time $t$. Thus, we have:

$$
\begin{aligned}
\sqrt{1-\bar{\alpha}_t}||\nabla_{\mathbf{x}_t} \log E_t(\mathbf{x}_t, \mathbf{y}) - \nabla_{\mathbf{x}_t} \log R_t(\mathbf{x}_t, \mathbf{y})|| &= \sqrt{1-\bar{\alpha}_t}||\nabla_{\mathbf{x}_t} \log R_0(\hat{\mathbf{x}}_0, \mathbf{y}) - \nabla_{\mathbf{x}_t} \log R_t(\mathbf{x}_t, \mathbf{y})|| \\
&\leq \sqrt{1-\bar{\alpha}_t}||\nabla_{\mathbf{x}_t} \log R_0(\hat{\mathbf{x}}_0, \mathbf{y}) - \nabla_{\hat{\mathbf{x}}_0} \log R_0(\hat{\mathbf{x}}_0, \mathbf{y})|| \\
&\quad + \sqrt{1-\bar{\alpha}_t}||\nabla_{\hat{\mathbf{x}}_0} \log R_0(\hat{\mathbf{x}}_0, \mathbf{y}) - \nabla_{\mathbf{x}_t} \log R_0(\mathbf{x}_t, \mathbf{y})|| \\
&\quad + \sqrt{1-\bar{\alpha}_t}||\nabla_{\mathbf{x}_t} \log R_0(\mathbf{x}_t, \mathbf{y}) - \nabla_{\mathbf{x}_t} \log R_t(\mathbf{x}_t, \mathbf{y})|| .
\end{aligned}
\tag{32}
$$

Here we take CFG as an example, while similar proof can be adopted for AutoG and any CFG-variant guidance methodologies. For clarity, using the score function formula Eq. (7) with the reward $R_t$ in CG shown in Eq. (9), we can derive:

$$
-\sqrt{1-\bar{\alpha}_t} \cdot \nabla_{\mathbf{a}} \log R_t(\mathbf{a}, \mathbf{y}) = w \left( \boldsymbol{\epsilon}_\theta(\mathbf{a}, t, \mathbf{y}) - \boldsymbol{\epsilon}_\theta(\mathbf{a}, t) \right) .
\tag{33}
$$

Now let us deal with the right hand side of Eq. (32) one by one:

$$
\begin{aligned}
\text{RHS Term 1} &= \sqrt{1-\bar{\alpha}_t}||\nabla_{\hat{\mathbf{x}}_0} \log R_0(\hat{\mathbf{x}}_0, \mathbf{y}) \cdot \frac{d\hat{\mathbf{x}}_0}{d\mathbf{x}_t} - \nabla_{\hat{\mathbf{x}}_0} \log R_0(\mathbf{x}_0, \mathbf{y})|| &&\text{(Chain rule)} \\
&\leq \sqrt{1-\bar{\alpha}_t}||\frac{d\hat{\mathbf{x}}_0}{d\mathbf{x}_t} - \mathbf{I}|| \cdot ||\nabla_{\hat{\mathbf{x}}_0} \log R_0(\hat{\mathbf{x}}_0, \mathbf{y})|| &&\text{(Extract the same term)} \\
&\leq w||\frac{d\hat{\mathbf{x}}_0}{d\mathbf{x}_t} - \mathbf{I}|| \cdot ||\boldsymbol{\epsilon}_\theta(\hat{\mathbf{x}}_0, 0, \mathbf{y}) - \boldsymbol{\epsilon}_\theta(\hat{\mathbf{x}}_0, 0)|| &&\text{(Use Eq. (33))} \\
&\leq 2wB||\frac{d\hat{\mathbf{x}}_0}{d\mathbf{x}_t} - \mathbf{I}|| , &&(\boldsymbol{\epsilon}_{\theta,t} \text{ is bounded by } B) \\
\text{RHS Term 2} &= w \cdot ||\boldsymbol{\epsilon}_\theta(\hat{\mathbf{x}}_0, 0, \mathbf{y}) - \boldsymbol{\epsilon}_\theta(\hat{\mathbf{x}}_0, 0) - \boldsymbol{\epsilon}_\theta(\mathbf{x}_t, 0, \mathbf{y}) + \boldsymbol{\epsilon}_\theta(\mathbf{x}_t, 0)|| &&\text{(Use Eq. (33))} \\
&\leq w \cdot ||\boldsymbol{\epsilon}_\theta(\hat{\mathbf{x}}_0, 0, \mathbf{y}) - \boldsymbol{\epsilon}_\theta(\mathbf{x}_t, 0, \mathbf{y})|| + w \cdot ||\boldsymbol{\epsilon}_\theta(\mathbf{x}_t, 0) - \boldsymbol{\epsilon}_\theta(\hat{\mathbf{x}}_0, 0)|| &&\text{(Triangular inequality)} \\
&\leq 2wL||\hat{\mathbf{x}}_0 - \mathbf{x}_t|| , &&(\boldsymbol{\epsilon}_{\theta,t} \text{ is Lipschitz continuous}) \\
\text{RHS Term 3} &= w \cdot ||\boldsymbol{\epsilon}_\theta(\mathbf{x}_t, 0, \mathbf{y}) - \boldsymbol{\epsilon}_\theta(\mathbf{x}_t, 0) - \boldsymbol{\epsilon}_\theta(\mathbf{x}_t, t, \mathbf{y}) + \boldsymbol{\epsilon}_\theta(\mathbf{x}_t, t)|| &&\text{(Use Eq. (33))} \\
&\leq w \cdot ||\boldsymbol{\epsilon}_\theta(\mathbf{x}_t, 0, \mathbf{y}) - \boldsymbol{\epsilon}_\theta(\mathbf{x}_t, t, \mathbf{y})|| + w \cdot ||\boldsymbol{\epsilon}_\theta(\mathbf{x}_t, t) - \boldsymbol{\epsilon}_\theta(\mathbf{x}_t, 0)|| &&\text{(Triangular inequality)} \\
&\leq 2wLt . &&(\boldsymbol{\epsilon}_{\theta,t} \text{ is Lipschitz continuous})
\end{aligned}
\tag{34}
$$

Combining them together, we obtain:

$$
\sqrt{1-\bar{\alpha}_t}||\nabla_{\mathbf{x}_t} \log E_t(\mathbf{x}_t, \mathbf{y}) - \nabla_{\mathbf{x}_t} \log R_t(\mathbf{x}_t, \mathbf{y})|| \leq 2wB||\frac{d\hat{\mathbf{x}}_0}{d\mathbf{x}_t} - \mathbf{I}|| + 2wL||\hat{\mathbf{x}}_0 - \mathbf{x}_t|| + 2wLt . \quad (\star)
\tag{35}
$$

If we denote $\Delta = \hat{\mathbf{x}}_0 - \mathbf{x}_t$, then we obtain the bound shown in Eq. (18) in the main text. When the one-step prediction $\hat{\mathbf{x}}_0 = \frac{1}{\sqrt{\bar{\alpha}_t}}(\mathbf{x}_t - \sqrt{1-\bar{\alpha}_t}\boldsymbol{\epsilon}_\theta(\mathbf{x}_t, t, \mathbf{y}))$ works well (e.g., at the end of denoising, $t$ is close to 0), we can further simplify the upper bound $(\star)$:

$$
\begin{aligned}
(\star) &= \frac{2wB}{\sqrt{\bar{\alpha}_t}}||(1-\sqrt{\bar{\alpha}_t})\mathbf{I} - \sqrt{1-\bar{\alpha}_t}\frac{\partial\boldsymbol{\epsilon}_\theta(\mathbf{x}_t, t, \mathbf{y})}{\partial\mathbf{x}_t}|| + \frac{2wL}{\sqrt{\bar{\alpha}_t}}||(1-\sqrt{\bar{\alpha}_t})\mathbf{x}_t - \sqrt{1-\bar{\alpha}_t}\boldsymbol{\epsilon}_\theta(\mathbf{x}_t, t, \mathbf{y})|| + 2wLt \\
&\leq \frac{2wB}{\sqrt{\bar{\alpha}_t}}(1-\sqrt{\bar{\alpha}_t} + \sqrt{1-\bar{\alpha}_t}||\frac{\partial\boldsymbol{\epsilon}_\theta(\mathbf{x}_t, t, \mathbf{y})}{\partial\mathbf{x}_t}||) + \frac{2wL}{\sqrt{\bar{\alpha}_t}}[(1-\sqrt{\bar{\alpha}_t})||\mathbf{x}_t|| + \sqrt{1-\bar{\alpha}_t}||\boldsymbol{\epsilon}_\theta(\mathbf{x}_t, t, \mathbf{y})||] + 2wLt \\
&\leq \frac{2wB}{\sqrt{\bar{\alpha}_t}}(1-\sqrt{\bar{\alpha}_t} + \sqrt{1-\bar{\alpha}_t}L) + \frac{2wL}{\sqrt{\bar{\alpha}_t}}[(1-\sqrt{\bar{\alpha}_t})||\mathbf{x}_t|| + \sqrt{1-\bar{\alpha}_t}B] + 2wLt \\
&= \underbrace{\frac{2wL}{\sqrt{\bar{\alpha}_t}}(1-\sqrt{\bar{\alpha}_t})}_{C_1}||\mathbf{x}_t|| + \underbrace{\frac{2wB}{\sqrt{\bar{\alpha}_t}}(1-\sqrt{\bar{\alpha}_t}) + \frac{4wBL}{\sqrt{\bar{\alpha}_t}}\sqrt{1-\bar{\alpha}_t} + 2wLt}_{C_2} .
\end{aligned}
\tag{36}
$$

By taking expectation of the above inequality with respect to $p_\theta(\mathbf{x}_t|\mathbf{y})$, we obtain:

$$
\begin{aligned}
\mathbb{E}_{p_\theta(\mathbf{x}_t|\mathbf{y})}[||\bar{\boldsymbol{\epsilon}}_{\theta,t}^\star - \bar{\boldsymbol{\epsilon}}_{\theta,t}||] &= \mathbb{E}_{p_\theta(\mathbf{x}_t|\mathbf{y})}[\sqrt{1-\bar{\alpha}_t}||\nabla_{\mathbf{x}_t} \log E_t(\mathbf{x}_t, \mathbf{y}) - \nabla_{\mathbf{x}_t} \log R_t(\mathbf{x}_t, \mathbf{y})||] \\
&\leq \mathbb{E}_{p_\theta(\mathbf{x}_t|\mathbf{y})}[C_1||\mathbf{x}_t|| + C_2] \\
&= C_1 \cdot \mathbb{E}_{p_\theta(\mathbf{x}_t|\mathbf{y})}[||\mathbf{x}_t||] + C_2 .
\end{aligned}
\tag{37}
$$

**Proof of Theorem 4.3.** Using the definition of $\bar{\epsilon}_{\theta,t}$ and $\bar{\epsilon}^{\star}_{\theta,t}$ given in Eqs. (10) and (17), we obtain:

$$\mathbb{E}_{p_\theta(\mathbf{x}_t|\mathbf{y})}[\bar{\epsilon}^{\star}_{\theta,t} - \bar{\epsilon}_{\theta,t}] = \sqrt{1-\bar{\alpha}_t} \cdot \mathbb{E}_{p_\theta(\mathbf{x}_t|\mathbf{y})}[\nabla_{\mathbf{x}_t} \log \frac{R_t(\mathbf{x}_t, \mathbf{y})}{E_t(\mathbf{x}_t, \mathbf{y})}]. \tag{38}$$

Now let us simplify the right-hand side. To begin with, we notice that chain rule states:

$$\int \frac{\partial p_\theta(\mathbf{x}_t|\mathbf{y})}{\partial \mathbf{x}_t} \log \frac{R_t(\mathbf{x}_t, \mathbf{y})}{E_t(\mathbf{x}_t, \mathbf{y})} \, d\mathbf{x}_t + \int p_\theta(\mathbf{x}_t|\mathbf{y}) \frac{\partial \log \frac{R_t(\mathbf{x}_t,\mathbf{y})}{E_t(\mathbf{x}_t,\mathbf{y})}}{\partial \mathbf{x}_t} \, d\mathbf{x}_t = \frac{\partial}{\partial \mathbf{x}_t} \int p_\theta(\mathbf{x}_t|\mathbf{y}) \log \frac{R_t(\mathbf{x}_t, \mathbf{y})}{E_t(\mathbf{x}_t, \mathbf{y})} \, d\mathbf{x}_t. \tag{39}$$

Moreover, the right-hand side of the above equation actually integrates out the variable $\mathbf{x}_t$, leaving it dependent solely on $\mathbf{y}$. Consequently, the derivative with respect to $\mathbf{x}_t$ becomes zero, and Eq. (39) simplifies to:

$$\begin{aligned}
&\int \frac{\partial p_\theta(\mathbf{x}_t|\mathbf{y})}{\partial \mathbf{x}_t} \log \frac{R_t(\mathbf{x}_t, \mathbf{y})}{E_t(\mathbf{x}_t, \mathbf{y})} \, d\mathbf{x}_t + \mathbb{E}_{p_\theta(\mathbf{x}_t|\mathbf{y})}[\nabla_{\mathbf{x}_t} \log \frac{R_t(\mathbf{x}_t, \mathbf{y})}{E_t(\mathbf{x}_t, \mathbf{y})}] = 0 \,, \\
&\Rightarrow \mathbb{E}_{p_\theta(\mathbf{x}_t|\mathbf{y})}[\nabla_{\mathbf{x}_t} \log \frac{R_t(\mathbf{x}_t, \mathbf{y})}{E_t(\mathbf{x}_t, \mathbf{y})}] = -\int p_\theta(\mathbf{x}_t|\mathbf{y}) \cdot \nabla_{\mathbf{x}_t} \log p_\theta(\mathbf{x}_t|\mathbf{y}) \cdot \log \frac{R_t(\mathbf{x}_t, \mathbf{y})}{E_t(\mathbf{x}_t, \mathbf{y})} \, d\mathbf{x}_t \,, \\
&\Rightarrow \mathbb{E}_{p_\theta(\mathbf{x}_t|\mathbf{y})}[\nabla_{\mathbf{x}_t} \log \frac{R_t(\mathbf{x}_t, \mathbf{y})}{E_t(\mathbf{x}_t, \mathbf{y})}] = -\mathbb{E}_{p_\theta(\mathbf{x}_t|\mathbf{y})}[\nabla_{\mathbf{x}_t} \log p_\theta(\mathbf{x}_t|\mathbf{y}) \cdot \log \frac{R_t(\mathbf{x}_t, \mathbf{y})}{E_t(\mathbf{x}_t, \mathbf{y})}] \,, \\
&\Rightarrow \mathbb{E}_{p_\theta(\mathbf{x}_t|\mathbf{y})}[\nabla_{\mathbf{x}_t} \log \frac{R_t(\mathbf{x}_t, \mathbf{y})}{E_t(\mathbf{x}_t, \mathbf{y})}] = -\mathbb{E}_{p_\theta(\mathbf{x}_t|\mathbf{y})}[-\frac{\epsilon_\theta(\mathbf{x}_t, t, \mathbf{y})}{\sqrt{1-\bar{\alpha}_t}} \cdot \log \frac{R_t(\mathbf{x}_t, \mathbf{y})}{E_t(\mathbf{x}_t, \mathbf{y})}] \,.
\end{aligned} \tag{40}$$

Thus, combining this equation with Eq. (38), we obtain:

$$\mathbb{E}_{p_\theta(\mathbf{x}_t|\mathbf{y})}[\bar{\epsilon}^{\star}_{\theta,t} - \bar{\epsilon}_{\theta,t}] = \mathbb{E}_{p_\theta(\mathbf{x}_t|\mathbf{y})}[\epsilon_{\theta,t} \cdot \log \frac{R_t(\mathbf{x}_t, \mathbf{y})}{E_t(\mathbf{x}_t, \mathbf{y})}]. \tag{41}$$

## E. Experimental Settings and Additional Numerical Results

### E.1. 1D and 2D Synthetic Examples

**1D Synthetic Example.** The 1D synthetic example is inspired by the work of Kynkäänniemi et al. (2024). We generate a total of 8,000 samples from the conditional distribution $q(x|y=0) = 0.5 \times \mathcal{N}(x|0.5, 0.25^2) + 0.5 \times \mathcal{N}(x|1.5, 0.25^2)$, and 8,000 samples from the unconditional data distribution $q(x) = 0.5 \times \mathcal{N}(x|-1, 0.5^2) + 0.5 \times \mathcal{N}(x|1, 0.5^2)$. For learning the conditional and unconditional distributions, we define separate noise prediction networks, $\epsilon_\theta(x_t, t, y)$ and $\epsilon_\theta(x_t, t)$. It is possible to follow the convention in CFG (Ho & Salimans, 2022) by using a shared-weight noise prediction network with label dropout during training, and we find no significant difference compared to our implementation in this specific case. We emphasize that this example only considers a single class with label $y = 0$, and, similar to (Kynkäänniemi et al., 2024), we regard the total number of classes and the distribution of other classes as irrelevant.

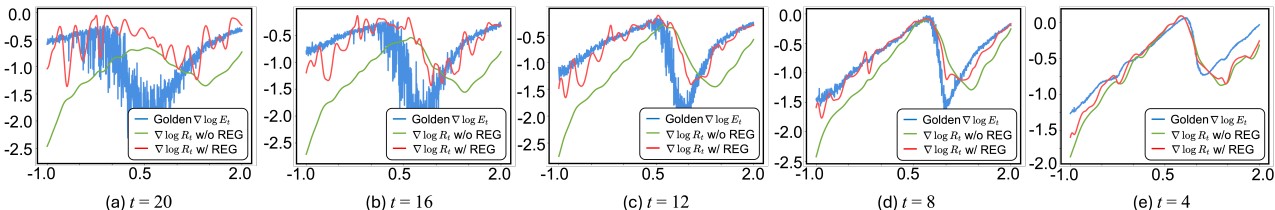

*Figure 5.* Guidance values are plotted along the X-axis in the range $[-1.0, 2.0]$ at different time steps.

In our noise prediction networks, we employ sinusoidal embeddings for time, class labels, and the coordinate input, each with a dimension of 128. These embeddings are concatenated and passed through an MLP with three hidden layers, each having 128 hidden units. We use 20 time steps with $\beta$ linearly scheduled from 0.001 to 0.2 (i.e., $\alpha_t$ is linear from $\alpha_1 = 1 - 0.001$ to $\alpha_{25} = 1 - 0.2$ in the DDPM notation). Note that for experimental purposes, we deliberately limit the capacity of the diffusion model by reducing the number of time steps since diffusion models with sufficient capacity can nearly perfectly learn these examples without any guidance. The network is trained using AdamW with a learning rate of 0.001 for 200

epochs. To evaluate $\nabla_{x_t} \log E_t(x_t, y)$ at a specific time step $t$, we follow the denoising process to generate 100 instances of $x_0$ given observing $x_t$ at $t$, and then evaluate the mean of their reward, according to Eq. (14).

We have plotted comparisons of the golden $\nabla \log E_t$, $\nabla \log R_t$ with REG, and $\nabla \log R_t$ without REG at various time steps in Figure 5 to complement Figure 1 in the main text. The fitting results are further shown in Figure 6.

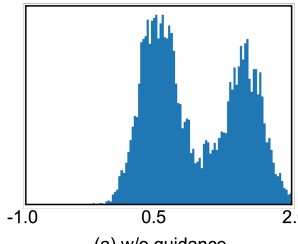 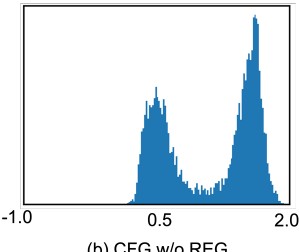 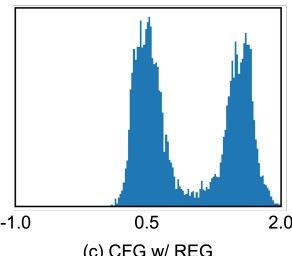

(a) w/o guidance        (b) CFG w/o REG        (c) CFG w/ REG

*Figure 6.* Histograms of 8000 samples drawn from a trained simple diffusion model are shown for three cases: (a) without any guidance, (b) using the vanilla CFG without REG, and (c) using the vanilla CFG with our proposed REG. The target conditional data distribution is $0.5 \times \mathcal{N}(0.5, 0.25^2) + 0.5 \times \mathcal{N}(1.5, 0.25^2)$. While all approaches correctly recover the mean locations, our method achieves better alignment of the peak heights, more accurately reflecting the target's equal magnitude Gaussian components.

**2D Synthetic Example.** We are inspired by Pärnamaa (2023) and design the 2D synthetic example as follows: We prepare the target shape (e.g. dinosaur) and extract the 2D coordinates from its SVG file. Then, we define a Gaussian mixture with equal weight coefficients as the data generation distribution, with the mean of Gaussian mixture component being every extracted grid coordinate, and the standard deviation being 0.05. This setup makes sure we can evaluate the log probability for a given $\mathbf{x}_0$. In this example, we define a single noise prediction network with label dropout (dropout probability of 0.2) during training, following the convention of CFG (Ho & Salimans, 2022). Similar to the 1D example, we use sinusoidal embeddings for time, class labels, and the coordinate input, each with a dimension of 512. These embeddings are concatenated and passed through an MLP with three hidden layers, each containing 128 units. We use 25 time steps with $\beta$ linearly scheduled from 0.001 to 0.2 (i.e., $\alpha_t$ transitions linearly from $\alpha_1 = 1 - 0.001$ to $\alpha_{25} = 1 - 0.2$ in the DDPM notation). The diffusion model is trained using the AdamW optimizer with a learning rate of 0.0001 for 200 epochs.

Evaluating the golden $\nabla \log E_t$ is performed via numerical integration using 500 samples. Empirically, we observe that unusually large values of $\nabla \log E_t$ occasionally appear. Increasing the number of samples in the numerical integration (e.g., from 100 to 500) helps mitigate this issue, as it provides a more accurate approximation of the expectation in higher-dimensional spaces. However, using 500 samples is already computationally expensive, as it requires not only evaluating $\log E_t$ but also computing its gradients through backpropagation. Thus, we exclude samples with abnormally large $\nabla \log E_t$ values when plotting Figure 2 as a straightforward remedy.

Since $\nabla \log E_t$ in this case represents the derivative of a scalar with respect to a 2D vector, it can be visualized as a gradient arrow in a 2D plane. However, as shown in Table 1, while the proposed REG consistently provides a better approximation of the optimal solution, this improvement is barely visible in columns (d)-(f) of Figure 2.

**E.2. Quantitative Results: Image Generation**

We have summarized the models used in our experiments in Table 2. In the main text, REG is derived using the $\epsilon$-prediction parametrization, where we simplify the Jacobian matrix by using $\hat{\mathbf{x}}_0 = \frac{1}{\sqrt{\bar{\alpha}_t}}(\mathbf{x}_t - \sqrt{1 - \bar{\alpha}_t}\epsilon_{\theta,t})$, leading to Eq. (21). For EDM2 (Karras et al., 2024b), which uses the $\mathbf{x}_0$ parametrization, the REG correction term trivially becomes $\partial(\mathbf{1}^T \cdot D_\theta(\mathbf{x}_t, t, y))/\partial \mathbf{x}_t$, where $D_\theta$ represents the $\mathbf{x}_0$-prediction network as defined in EDM2 (Karras et al., 2022; 2024b). Additionally, SD-v1-4 and SD-XL employ the PNDM sampler (Liu et al., 2022) and the Euler Discrete sampler, respectively, which differ from DDPM. Nevertheless, we use Eq. (21) in these cases as well, and it performs effectively in practice.

We conduct our experiments using the open-source DiT (Peebles & Xie, 2023), EDM2 (Karras et al., 2024b), and Huggingface Diffusers codebases, modifying their source code to support various guidance techniques. The interval CFG (Kynkäänniemi et al., 2024) study provides the interval ranges for DiT and EDM2, which we adopt here for consistency. For the "bad" models used in AutoG, we rely on the publicly available checkpoints shared by Karras et al. (2024a). However, when the interval range or the "bad" model checkpoint is unavailable (e.g., in text-to-image tasks), we cannot reproduce the

interval CFG and AutoG baselines. This limitation arises because interval CFG requires an exhaustive hyper-parameter search to identify the interval range, and AutoG relies on a carefully designed "bad" model. For linear CFG, we set the guidance weight to zero at the start of denoising and gradually increase the weight scale toward the end. For cosine CFG (Gao et al., 2023), we use a power of 2 for class-conditional ImageNet generation and a power of 1 for text-to-image generation, as we find that a power of 2 produces extremely poor results in text-to-image generation. In class-conditional ImageNet generation, we evaluate FID and IS metrics using 50,000 generated images, following the protocols outlined in DiT (Peebles & Xie, 2023) and EDM2 (Karras et al., 2024b). For text-to-image generation, we randomly select one caption per image from the COCO-2017 validation dataset, creating 5,000 pairs of images and captions. FID and CLIP scores are evaluated using TorchMetrics.

Additional Pareto front results for class-conditional ImageNet generation are presented in Figure 7. Qualitative examples of generated images corresponding to various text prompts in the text-to-image generation task are illustrated in Figures 8–10.

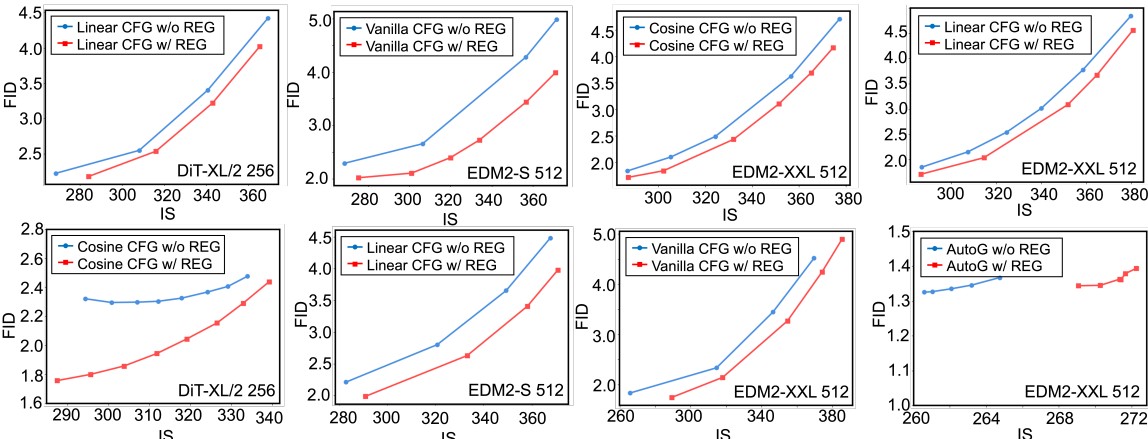

Figure 7. Additional results for the Pareto front of FID versus IS are shown by varying the guidance weight $w$ over a broad range for different methods. Curves positioned further toward the bottom-right indicate superior performance.

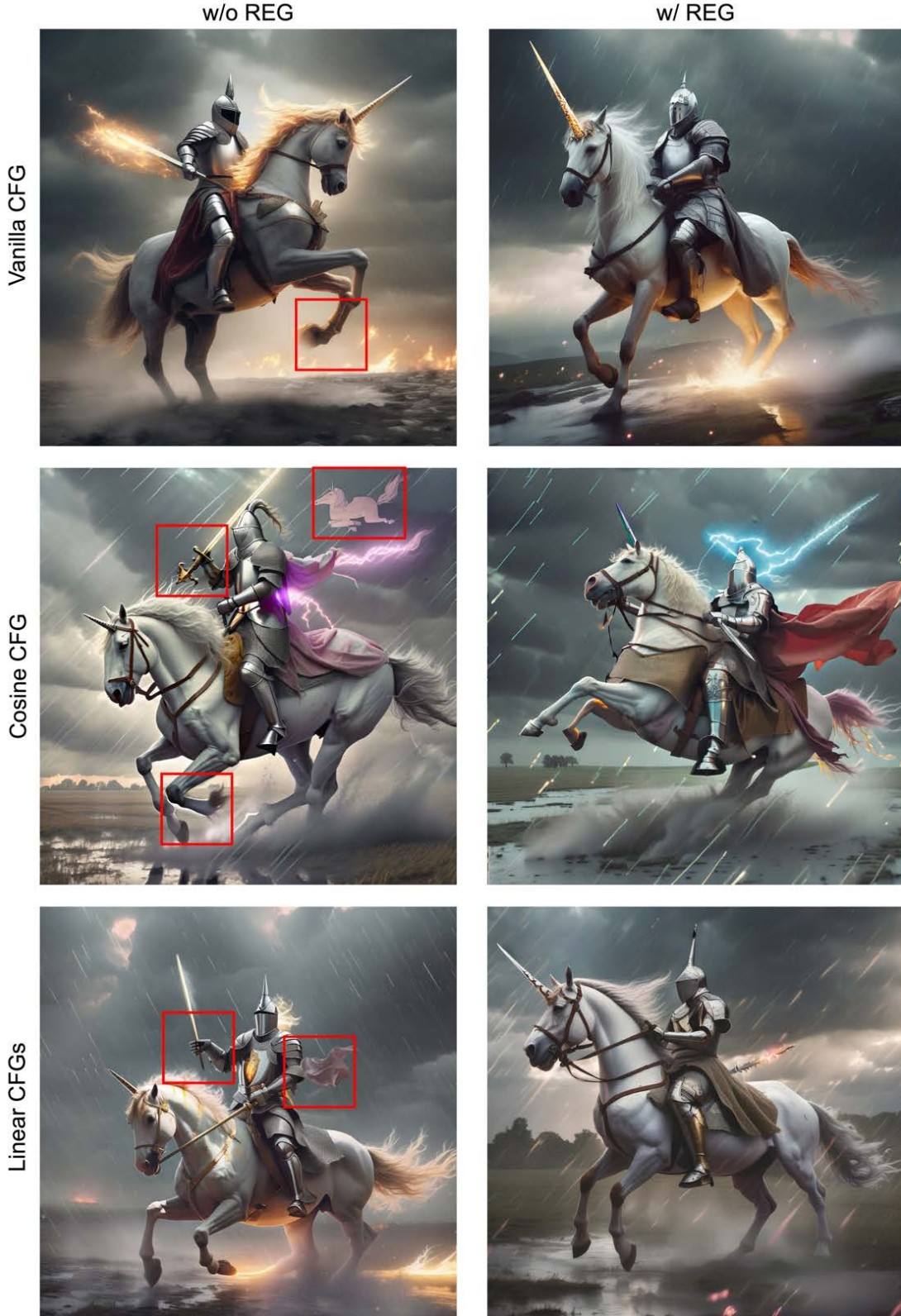

*Figure 8.* Generated images by different methods are shown under high guidance strength given the prompt "A medieval knight riding a glowing unicorn through a stormy battlefield".

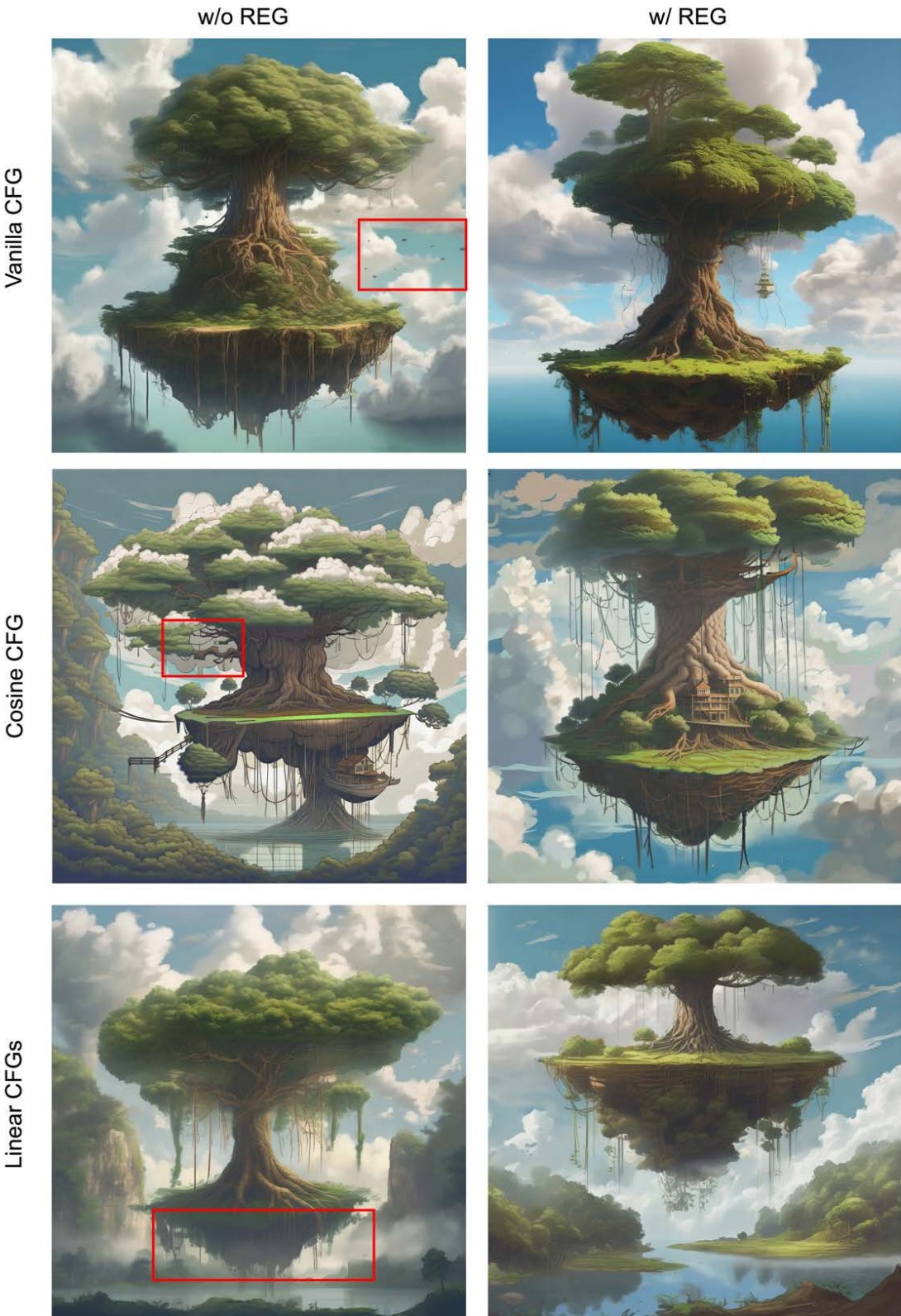

*Figure 9.* Generated images by different methods are shown under high guidance strength given the prompt "A floating island with a giant tree whose roots hang down into the clouds".

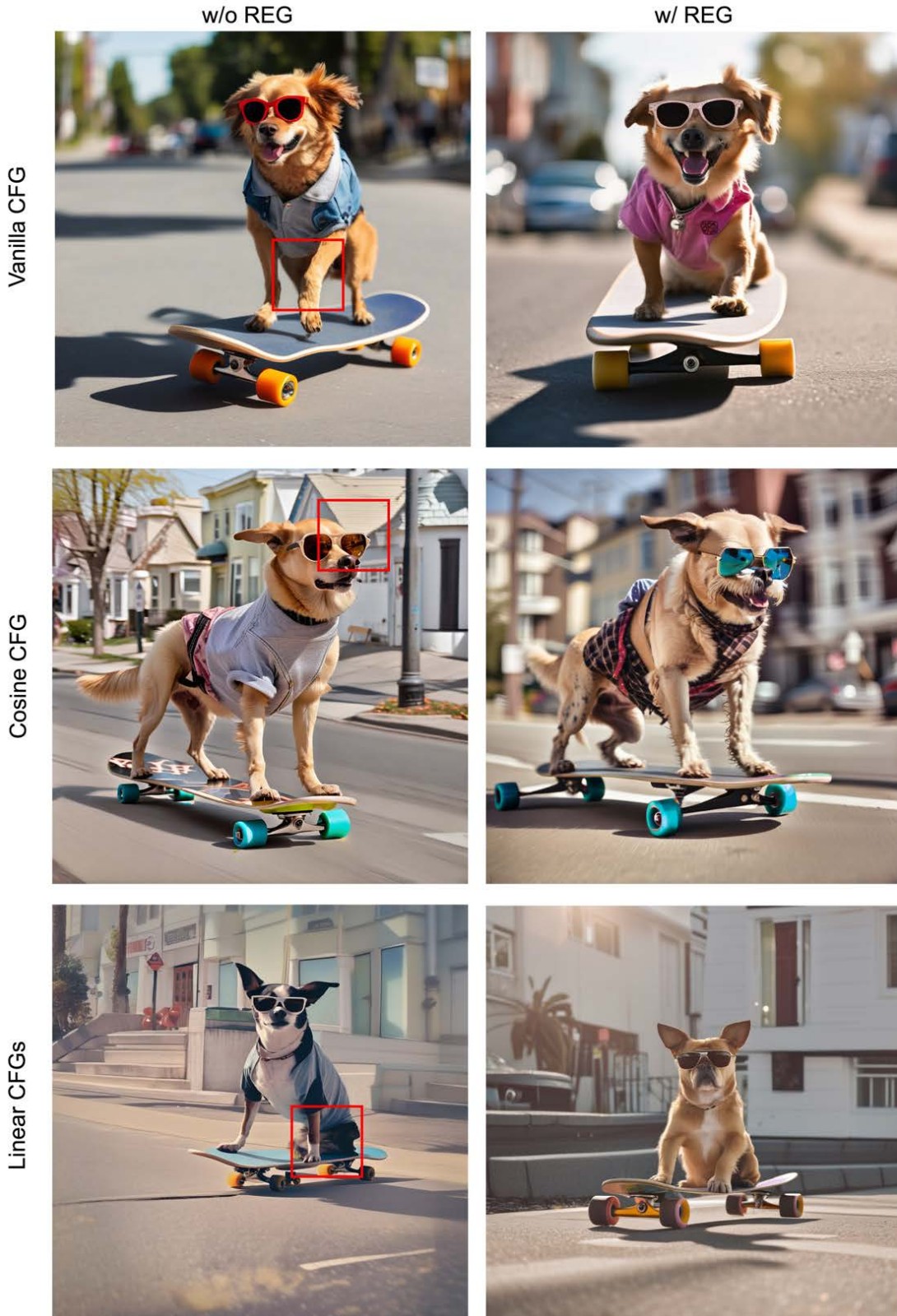

*Figure 10.* Generated images by different methods are shown under high guidance strength given the prompt "A dog wearing sunglasses and riding a skateboard on a sunny street".

