# OpenReview forum: "REG: Rectified Gradient Guidance for Conditional Diffusion Models"
_ICML.cc/2025/Conference — ICML 2025 poster_

### Official Review · Reviewer_U91Q · 2025-03-10

**Overall Recommendation:** 4

**Summary:**

This paper studies the foundations of classifier-free guidance (CFG). It finds that the common interpretation of scaling marginal distribution $p(x_t \vert y)$, i.e., use guidance to change sampling trajectories,  is not theoretically grounded as it is impossible to construct a DDPM process that corresponds to the scaled distribution. The paper then proposes to scale the joint distribution $p(x_{0:T} \vert y)$, which could be proved to produce a valid DDPM process. With this new framework, CFG can be treated as a special approximation

The paper then proposes a new form of approximation to the theoretically-grounded formulation, i.e., REG, and demonstrates its effectiveness on 1D and 2D synthetic data as well as high-dimension image synthesis tasks, i.e., ImageNet generation.

**Claims And Evidence:**

The paper is well-written and the claims are well-supported.

I have one question about Sec. 3's "Common Interpretation of Guidance": I do not think that the original CG or CFG only scales the marginal distribution of $p(x_0 \vert y)$. However, the authors state
> (L161 left) CG and CFG rewards are explicitly stated in (Ho & Salimans, 2022)

Can authors point me to the specific part where **marginal distribution** scaling is mentioned in CFG paper?

**Essential References Not Discussed:**

N/A

**Experimental Designs Or Analyses:**

The experiments are extensive and convincing.

**Methods And Evaluation Criteria:**

The evaluations are solid and thorough.

**Other Comments Or Suggestions:**

N/A

**Other Strengths And Weaknesses:**

The paper is solid in theory and thorough in experiments. I do not find a major weakness in the paper besides my questions spread in other sections.

**Questions For Authors:**

People have applied CFG to flow-matching-based approaches by modifying the velocity functions [a] and the velocity has close connections to score functions. Thus I am wondering whether REG could be easily applied to a model based on flow matching.

1. If the answer is affirmative, I am wondering whether the authors have the resources to apply REG to models based on flow matching, e.g., Stable Diffusion 3. This would further enhance the impact of the paper.
2. If the answer is negative, what would be the difficulty?

[a] Ma et al., SiT: Exploring Flow and Diffusion-based Generative Models with Scalable Interpolant Transformers. ECCV 2024.

**Relation To Broader Scientific Literature:**

The paper falls in the recent community effort to fundamentally understand the mechanism of CFG.

**Theoretical Claims:**

I read through all the proofs and found they are detailed and easy to follow.

---

> ### Author Rebuttal · Authors · 2025-03-30
>
> Thank you so much for acknowledging our contribution and the constructive feedback. Below we address each concern raised.
>
> ---
>
> **Q1.**  Where marginal distribution scaling is mentioned in CFG paper?
>
> **A1.**  Thank you for the great question. As a quick recap, Section II of our paper explains the theoretical pitfall of guidance. We begin by examining the case of scaling only the terminal marginal distribution (Eq. (5)) and then generalize to scaling all marginal distributions (Eq. (8)). The latter is a stricter goal that subsumes the former. We show that neither is theoretically well-justified.
>
> Strictly speaking, Ho & Salimans (2022) state that classifier guidance (CG) leads to sampling from scaled versions of all marginal distributions—see the second equation under Algorithm 1 on page 4. This aligns with our Eq. (8)–(9), where their reward term $p_\theta(c | z_\lambda)^w$ corresponds to our notation $R_t(x_t, y) = p_\phi(y | x_t)^w$ in Eq. (9). They also express the CFG reward using $p^i(c | z_\lambda)$ in the second-to-last paragraph before Section 4. Thus, the CG and CFG reward forms are indeed explictly stated there.
>
> For presentation clarity, we choose to start with the simpler case of scaling only the terminal marginal (Eq. (5)), but we acknowledge that Eq. (8) is the original formulation in Ho & Salimans (2022).  We will revise the text and footnote accordingly to make our claim more accurate.
>
> ---
>
> **Q2.** The qualitative visualizations in Fig. 8-10.
>
> **A2.** Thanks for the great question. To elaborate: in Figure 8, the knight’s gesture while holding the sword appears unnatural, and the horse’s legs are blurry in the baseline images. In Figure 9, there are black artifacts in the sky (not birds), and the lower part of the tree suffers from low contrast in the baseline images. In Figure 10, the dog’s paws are unnatural, and the sunglasses are deformed in the baseline images.
>
> We agree with the reviewer that these differences might be subtle. For additional context, it might be helpful to consult qualitative results reported in other works to put our qualitative results in context --- see, for example, Figure 6-7 of [1], Figure 4 of [2], Figure 1 of [3], and [4].
>
> To better address the concern, we have followed the suggestions and added multiple samples with the same method for a given prompt. These can be found in File 1 and File 2 at this [anonymous link](https://anonymous.4open.science/r/icml-1655-rebuttal).
>
> Please note that due to OpenReview rebuttal format constraints, we are unable to include figures directly in this response. However, anonymous links are permitted under ICML guidelines, and we will include these visualizations in the revised manuscript.
>
> ---
>
> **Q3.** Apply REG to flow-matching based diffusion models.
>
> **A3.** Thank you for the insightful question. We agree that evaluating REG in the context of flow-matching models would further enhance the impact of our work.
>
> Due to time and resource constraints—particularly the large model size (SDv3 has over 2 billion parameters) and its lengthy inference time—we were unable to run experiments on SD v3. However, we conduct extra experiments on SD v2.1, a velocity-parameterized diffusion model. This setup was mutually requested by other reviewers and, we believe, is also relevant to the current question. Namely, while velocity-parameterized models are not identical to flow-matching approaches, they are closely related. In fact, the distinction between them can often be seen as a subtle difference in the noise schedule [5]. We perform a grid search over guidance weights for different methods with SD v2.1 and report the best FID $\downarrow$ and CLIP $\uparrow$ scores for each method in the table below.
>
> We will perform thorough evaluations on REG for flow-matching based generative models and report them in the final version if the paper is accepted.
>
> |        | (CLIP, FID) w/ REG          | (CLIP, FID) w/o REG       |
> |-|-|-|
> | Linear CFG    | (31.62, 27.83)           | (31.40, 28.94)         |
> | Cosine CFG    | (31.48, 23.32)           | (31.72, 24.54)         |
>
> ---
>
> **References**
>
> [1] Tuomas Kynkäänniemi et al., 'Applying Guidance in a Limited Interval Improves Sample and Distribution Quality in Diffusion Models,' Neurips, 2024.
>
> [2] Tero Karras et al., 'Guiding a Diffusion Model with a Bad Version of Itself,' Neurips, 2024.
>
> [3] Tianwei Yin et al., 'One-step Diffusion with Distribution Matching Distillation,' CVPR, 2024.
>
> [4] Xi Wang et al., 'Analysis of Classifier-Free Guidance Weight Schedulers,' TMLR, 2024.
>
> [5] Ruiqi Gao et al., 'Diffusion Meets Flow Matching: Two Sides of the Same Coin', https://diffusionflow.github.io/.

---

> > ### Comment · Reviewer_U91Q · 2025-04-03
> >
> > I thank the authors for their time and effort in addressing my questions and concerns. I carefully read the other reviews and the rebuttal, I would like to keep my positive rating for this work for its solid theoretical principles and superior performance.

---

> > > ### Author Response · Authors · 2025-04-03
> > >
> > > We sincerely thank the reviewer for the positive feedback and for recognizing the value of our work. We will make further efforts to improve the clarity and quality of the paper according to the suggestions.

---

### Official Review · Reviewer_BraT · 2025-03-11

**Overall Recommendation:** 3

**Summary:**

This paper introduces a Rectified Gradient Guidance to improve conditional generation in diffusion models. Also the paper provides the theoretical foundation of the proposed guidance method. Experiments show the proposed method enhances the quality of generated images.

**Claims And Evidence:**

Yes

**Essential References Not Discussed:**

No

**Experimental Designs Or Analyses:**

Yes

**Methods And Evaluation Criteria:**

Yes

**Other Comments Or Suggestions:**

None

**Other Strengths And Weaknesses:**

Strengths:

1. The paper is well-written and easy to follow.

2. The paper provides the theoretical insight about the limitations in current guidance method and proposes a solution to improve the performance of existing guidance methods.

3. The authods conduct extensive experiments to justify the effectiveness of the proposed method.

Weaknesses:

1. The paper lacks reporting on inference time requirements. The authors should clarify whether the proposed REG introduces computational overhead compared to baseline methods.

2. The visualization results presented in the paper are insufficient to fully demonstrate the method's effectiveness. Additional qualitative examples across diverse scenarios would strengthen the paper.

3. For the 1D and 2D experiments, the evaluation would be more convincing if conducted on standard public datasets rather than custom data. The reported accuracy for 2D generation also appears quite low.

**Questions For Authors:**

1. In Table 3, why do most of the guidance methods exhibit worse performance than vanilla CFG, particularly in the context of the SD model?

2. Same problem also appear in the DiT-XL model.

**Relation To Broader Scientific Literature:**

The proposed Rectified Gradient Guidance is related to the guidance techniques in diffusion models, like classifier guidance and classifier-free guidance.

The paper replaces the scaled marginal distribution target with a valid scaled joint distribution objective, aligning with the theoretical motivations of guidance methods. This theoretical advancement builds on prior works that have explored the statistical theory and mathematical foundations of conditional diffusion models

**Theoretical Claims:**

Yes, the theorems in section 4.

---

> ### Author Rebuttal · Authors · 2025-03-30
>
> Thank you so much for acknowledging our contribution and the constructive feedback.
>
> ---
>
> **Q1.** Runtime and memory cost.
>
> **A1.** Thanks for the great question. The tables below summarize runtime and peak memory usage of CFG and REG on a single NVIDIA A40 GPU. Runtime is reported using example batch sizes, while memory is measured with batch size 1 to isolate per-image cost. Since REG introduces one extra gradient computation on top of vanilla CFG, a moderate increase in runtime and memory usage is expected. Similar inference-time gradient calculations have also been explored in Universal Guidance (Arpit Bansal et al., 2024), albeit in a different context.
>
> We emphasize that our main contribution lies in correcting CFG theory, and REG serves as an empirical validation. Its practical deployment depends on the specific application and acceptable overhead. We will include these tables in the updated paper.
>
> | Model | Resol. | Batch Size | CFG/REG Runtime (sec) | Increase (x) |
> |-|-|-|-|-|
> | EDM2-S | 64    | 8  | 25.96 / 42.99    | 1.66    |
> | DiT-XL/2 | 256 | 8   | 59.79 / 94.23    | 1.58    |
> | EDM2-S | 512   | 8  | 46.14 / 62.87    | 1.36   |
> | EDM2-XXL | 512 | 8   | 49.21 / 92.60   | 1.88    |
> | SD-V1.4 | 512  | 4   | 32.63 / 39.54  | 1.21  |
> | SD-V2.1 | 768  | 4   | 36.55 / 59.76   | 1.64  |
> | SD-XL | 1024   | 2 | 47.48 / 74.52    | 1.57   |
>
> | Model | Resol.   | CFG / REG GPU Peak Mem (GB) | Increase (x) |
> |-|-|-|-|
> | EDM2-S | 64       | 0.87 / 1.49        | 1.71       |
> | DiT-XL/2 | 256    | 4.15 / 5.01         | 1.21       |
> | EDM2-S | 512      | 1.19 / 1.81      | 1.52       |
> | EDM2-XXL | 512    | 4.59 / 7.31      | 1.59   |
> | SD-V1.4 | 512     | 2.73 / 4.39        | 1.61     |
> | SD-V2.1 | 768     | 2.72 / 6.51     | 2.39     |
> | SD-XL | 1024      | 6.91 / 19.49        | 2.82    |
>
> ---
>
> **Q2.** Extra visualization results.
>
> **A2.** Thanks for the constructive remark. We respectfully point out that Figures 1, 2, 5, 8, 9, and 10 already provide qualitative visualizations for synthetic 1D/2D cases and real benchmarks. To address this concern, extra visualizations (e.g., standard 2D toy dataset, text-to-image and class-conditioned generation) are performed and added in this [anonymous link](https://anonymous.4open.science/r/icml-1655-rebuttal).
>
> Please note that due to OpenReview rebuttal format constraints, we are unable to include figures directly in this response. However, anonymous links are permitted under ICML guidelines, and we will include these visualizations in the revised manuscript.
>
> ---
>
> **Q3.** Standard public 1D and 2D datasets and 2D generation accuracy.
>
> **A3.** Thanks for the helpful feedback. Our 1D setup follows standards from prior works, such as Interval Guidance (Kynkäänniemi et al., 2024) and Autoguidance (Karras et al., 2024). For 2D, we use custom shapes with fine-grained structures, more challenging than standard 2D toy datasets like “two moons” or “Swiss rolls.” Extra results on standard 2D datasets are performed and can be accessed via this [anonymous link](https://anonymous.4open.science/r/icml-1655-rebuttal) (anonymous links permitted by ICML guidelines).
>
> About generation accuracy, we clarify that Figure 2(b) appears reasonably well when compared to the target in Figure 2(a). We acknowledge that the results may not be perfect, as we use a relatively small diffusion model in order to compute the golden gradient $\nabla \log E_t$ efficiently. We also clarify that the metric in Table 1 indicates how often REG achieves lower error compared to no REG. A value x\%>50% suggests consistent improvement.
>
> ---
>
> **Q4.** Cosine and linear CFG perform worse than vanilla in SD models and DiT-XL/2.
>
> **A4.** Thanks for the great question. We first want to clarify that our experiments are designed to verify whether REG can enhance a given guidance method. This argument has been validated by the results shown in Table 2 and 3. Our aim is not to compare “cosine + REG” vs. “vanilla CFG,” since such a comparison conflates the base method’s performance with the effect of REG. Hence, we believe the essential question here is that cosine and linear CFG perform worse than vanilla CFG in SD models and DiT-XL/2 (they do perform better in EDM2).
>
> This trend is consistent with results shown in related works, such as the analysis of CFG weight schedulers in Xi Wang et al., TMLR 2024 --- Both our Figure 4 (right) and their Figure 7(c) show that vanilla CFG achieves the lowest FID, linear CFG shifts slightly upward, and cosine CFG shifts it further upward in CLIP-FID space.
>
> Two factors likely contribute: (i) all these methods are heuristic, and their effectiveness can vary across model architectures and datasets. (ii) linear and cosine CFG require tuning two hyperparameters, while vanilla CFG uses only one—making linear and cosine CFG more expensive and less robust to optimize.
>
> We will include this clarification in the updated paper.

---

### Official Review · Reviewer_vJrA · 2025-03-12

**Overall Recommendation:** 3

**Summary:**

This paper addresses the discrepancy between the theoretical motivation and practical implementation of guidance techniques in conditional diffusion models. The authors propose a new method called Rectified Gradient Guidance (REG) to improve the performance of existing guidance methods.

The main findings of the paper include the identification of a significant gap between the theoretical derivations and practical implementations of current guidance techniques. The authors demonstrate that the commonly used marginal scaling approach is theoretically invalid and propose a joint distribution scaling objective as a valid alternative.

The key algorithmic contribution is the introduction of REG, which incorporates a novel correction term into existing guidance methods. This correction term is derived from a theoretical analysis of the optimal guidance solution and is designed to better approximate this optimal solution under practical constraints.

The main results show that REG provides a better approximation to the optimal solution than prior guidance techniques in 1D and 2D experiments. Extensive experiments on class-conditional ImageNet and text-to-image generation tasks demonstrate that incorporating REG consistently improves Fréchet Inception Distance (FID) and Inception/CLIP scores across various settings compared to its absence.

The conceptual contribution lies in establishing a unified theoretical framework for understanding guidance techniques in conditional diffusion models. The authors theoretically prove the invalidity of marginal scaling and demonstrate that established guidance implementations are approximations to the optimal solution with quantified error bounds.

The practical contribution is that REG is shown to be compatible with various guidance techniques and diffusion model architectures. The method can be easily integrated into existing diffusion pipelines and consistently enhances performance without requiring significant computational overhead.

**Claims And Evidence:**

The claims made in the submission are supported by clear and convincing evidence. The authors provide theoretical analysis, mathematical derivations, and extensive experimental results to validate their claims.

1. The discrepancy between theoretical motivation and practical implementation of guidance techniques is demonstrated through detailed analysis of the marginal scaling approach and its limitations (Section 3). The authors show mathematically why marginal scaling is invalid and how it conflicts with the constraints of diffusion models.

2. The claim that established guidance implementations are approximations to the optimal solution is supported by Theorem 4.1, which establishes the optimal solution under joint scaling, and Theorems 4.2 and 4.3, which quantify the approximation error of current methods. The authors clearly show the gap between current practices and the optimal solution.

3. The effectiveness of REG in 1D and 2D synthetic examples is demonstrated through visual comparisons (Figure 1 and Figure 2) and quantitative win ratios (Table 1). These results clearly show that REG provides a better approximation to the optimal solution than previous methods.

4. The improvement in performance on class-conditional ImageNet and text-to-image generation tasks is supported by comprehensive quantitative results (Tables 2 and 3) across multiple model architectures and guidance techniques. The Pareto front analyses (Figures 3 and 4) further demonstrate the consistent improvement provided by REG.

5. The compatibility of REG with various guidance techniques and diffusion model architectures is shown through experiments with different models (DiT, EDM2, SD-v1-4, SD-XL) and guidance methods (vanilla CFG, cosine CFG, linear CFG, interval CFG, AutoG). The implementation details confirm that REG can be easily integrated into existing pipelines.

**Essential References Not Discussed:**

None

**Experimental Designs Or Analyses:**

The experimental designs and analyses in this paper are generally sound and valid, providing comprehensive support for the proposed REG method across different scenarios and applications.

For the 1D and 2D synthetic examples, the experimental design effectively demonstrates REG's improvement in a controlled setting where ground truth can be computed. The comparisons are fair and isolate the effect of the guidance method, with win ratios and visual comparisons providing clear evidence of REG's effectiveness.

In class-conditional ImageNet generation, the evaluation is comprehensive, testing across multiple resolutions and model architectures. The use of standard metrics like FID and IS is appropriate, and the Pareto front analysis effectively shows how REG improves the efficiency of guidance techniques. The consistent improvement across different models and guidance methods strengthens the validity of the claims.

For text-to-image generation on COCO-2017, the experimental design is appropriate, evaluating REG with different model architectures and relevant metrics. The qualitative examples complement the quantitative results, providing additional evidence of improved generation quality.

While the experimental designs are robust, there are minor areas where additional analysis could enhance the work. More detailed analysis of computational overhead, especially for larger models, would be beneficial. Additional ablation studies could better isolate the contributions of different components of REG. Including more diverse and challenging prompts in text-to-image generation would further demonstrate REG's robustness. Comparing REG against other recently proposed guidance enhancements would also be valuable.

**Methods And Evaluation Criteria:**

The proposed Rectified Gradient Guidance (REG) method and the evaluation criteria utilized in this paper are both well-suited for addressing the problem of improving guidance techniques in conditional diffusion models.

REG directly targets the identified theoretical-practical gap in existing guidance methods by introducing a theoretically justified correction term derived from optimal guidance solutions under joint distribution scaling. This approach is logical as it enhances established guidance techniques rather than replacing them entirely, ensuring compatibility with various existing methods and model architectures.

The evaluation criteria, including benchmark datasets like ImageNet and COCO, along with performance metrics such as FID, IS, and CLIP score, provide a comprehensive assessment of REG's effectiveness. The combination of synthetic and real-world datasets, together with both quantitative metrics and qualitative visual comparisons, ensures a thorough evaluation of the method's impact across different scenarios and applications.

**Other Comments Or Suggestions:**

1. In Section 3, when discussing the invalidity of marginal scaling, adding a brief intuitive explanation alongside the mathematical proofs might help readers better grasp the concept.

2.  In the experimental sections, providing a summary table that compares the performance of REG across different guidance methods and model architectures could help readers quickly see the consistent improvements.



3. Including more specific implementation details about the REG correction term, particularly regarding computational considerations, would be helpful for practitioners looking to implement the method.

4.  While the paper mentions computational overhead, a more detailed discussion of potential limitations, such as increased memory requirements or compatibility with certain model architectures, would provide a more complete picture.

**Other Strengths And Weaknesses:**

# Strengths

1. The paper demonstrates originality by identifying and addressing a fundamental theoretical-practical gap in guidance techniques for diffusion models. While classifier guidance and classifier-free guidance have become standard approaches, this work provides a novel theoretical framework that re-examines their foundations. The identification of the invalidity of marginal scaling and the proposal of joint distribution scaling represent creative advances that build upon but significantly extend previous work.

2. The improvements demonstrated by REG across multiple benchmarks and model architectures indicate substantial practical significance. For conditional diffusion models, which are widely used in applications like image generation, text-to-image synthesis, and video generation, even small improvements in guidance techniques can lead to meaningful enhancements in output quality and diversity. The versatility of REG, as shown in the experiments, suggests it could become a standard enhancement in future diffusion model implementations.

3. The paper is well-written and structured logically. The theoretical sections build upon each other in a coherent manner, making complex concepts accessible. The experimental results are presented clearly with appropriate visualizations and quantitative analyses. The appendices provide additional details that enhance reproducibility and understanding.

# Weaknesses

1. While the paper mentions that REG introduces minor computational overhead, a more detailed analysis of the additional computational requirements would be beneficial, especially for larger models and in production settings. This could help practitioners better understand the trade-offs when implementing REG.

2. The paper demonstrates REG's effectiveness across several model architectures and guidance methods, but a more extensive analysis of its performance across a broader range of architectures and applications would strengthen the claims of generalizability. Additionally, testing on more diverse and challenging datasets beyond ImageNet and COCO could provide further insight into its robustness.

3. The paper could benefit from a more comprehensive comparison against other recently proposed guidance enhancements. This would help establish REG's position relative to other state-of-the-art methods and highlight its unique advantages.

4. While REG is presented as a versatile enhancement, the implementation details, particularly regarding the correction term, might require careful tuning and understanding of the underlying diffusion framework. Providing more implementation guidance or open-source code could lower the barrier to adoption for practitioners.

**Questions For Authors:**

The paper mentions that REG introduces minor computational overhead but doesn't provide specific details. Could you quantify the additional computational requirements of REG compared to standard guidance methods, especially for larger models like SD-XL? This would help clarify practical trade-offs. If the overhead is substantial, it might affect the practical significance despite performance improvements.


The experiments demonstrate REG's effectiveness across several model architectures, but how does REG perform with score-based generative models that use different parameterizations or sampling schemes? Understanding its performance on architectures beyond those tested would clarify its generalizability.

**Relation To Broader Scientific Literature:**

REG can be seen as an evolution of classifier-free guidance (Ho & Salimans, 2022) with a theoretically motivated correction term. Unlike previous enhancement techniques like AutoG (Karras et al., 2024a), which requires identifying a "bad" model version, REG provides a general correction applicable to various guidance methods without additional model training or complex setup.

**Theoretical Claims:**

I've carefully examined the theoretical claims and  proofs presented in this paper, focusing on the key theoretical contributions that form the foundation of the proposed REG method.

The theoretical framework begins by identifying a critical issue in existing guidance techniques: the discrepancy between the theoretical motivation based on marginal distribution scaling and the practical implementation. The authors demonstrate that marginal scaling is theoretically invalid due to the constraints of the diffusion model's denoising process. This is established through a detailed analysis of the reverse denoising process and the implications of attempting to scale marginal distributions at different time steps.

The paper then introduces the concept of joint distribution scaling as a valid alternative. Theorem 4.1 is central to this argument, establishing the form of the optimal noise prediction network under joint scaling. The proof of this theorem is rigorous and follows logically from the definition of the scaled joint distribution objective. The authors demonstrate the existence and uniqueness of the transition kernels corresponding to the scaled joint distribution, and derive the form of the updated noise prediction network. This theorem provides the theoretical justification for the REG method.

Building on this foundation, Theorems 4.2 and 4.3 analyze the approximation error of existing guidance methods compared to the optimal solution. These proofs are technically sound and provide quantitative bounds on the approximation error. The analysis considers the practical constraints of guidance implementation and shows how the lack of future foresight affects the accuracy of guidance signals. The proofs involve careful application of mathematical analysis and probability theory, considering the properties of the diffusion process and the guidance rewards.

The derivation of the REG correction term is also well-supported theoretically. The authors start from the optimal guidance equation and make reasonable approximations to derive a practical correction term that can be implemented in existing diffusion frameworks. The chain rule application and Jacobian simplifications are justified, and the empirical validation in the experiments supports the effectiveness of these approximations.

---

> ### Author Rebuttal · Authors · 2025-03-30
>
> Thank you so much for acknowledging our contribution and the constructive feedback.
>
> ---
>
> **Q1.** Runtime and memory cost.
>
> **A1.** Thanks for the great question. The tables below summarize runtime and peak memory usage of CFG and REG on a single NVIDIA A40 GPU. Runtime is reported using example batch sizes, while memory is measured with batch size 1 to isolate per-image cost. Since REG introduces one extra gradient computation on top of vanilla CFG, a moderate increase in runtime and memory usage is expected. Similar inference-time gradient calculations have also been explored in Universal Guidance (Arpit Bansal et al., 2024), albeit in a different context.
>
> We emphasize that our main contribution lies in correcting the CFG theory, and REG serves as an empirical validation. Its practical deployment depends on the specific application and acceptable overhead. We will include these tables in the updated paper.
>
>
> | Model | Resol. | Batch Size | CFG/REG Runtime (sec) | Increase (x) |
> |-|-|-|-|-|
> | EDM2-S | 64    | 8          | 25.96 / 42.99        | 1.66     |
> | DiT-XL/2 | 256 | 8          | 59.79 / 94.23        | 1.58       |
> | EDM2-S | 512   | 8          | 46.14 / 62.87        | 1.36       |
> | EDM2-XXL | 512 | 8          | 49.21 / 92.60        | 1.88      |
> | SD-V1.4 | 512  | 4          | 32.63 / 39.54        | 1.21  |
> | SD-V2.1 | 768  | 4          | 36.55 / 59.76        | 1.64   |
> | SD-XL | 1024   | 2          | 47.48 / 74.52        | 1.57   |
>
> | Model | Resol.   | CFG / REG GPU Peak Mem (GB) | Increase (x) |
> |-|-|-|-|
> | EDM2-S | 64       | 0.87 / 1.49      | 1.71     |
> | DiT-XL/2 | 256    | 4.15 / 5.01      | 1.21    |
> | EDM2-S | 512      | 1.19 / 1.81    | 1.52       |
> | EDM2-XXL | 512    | 4.59 / 7.31     | 1.59    |
> | SD-V1.4 | 512     | 2.73 / 4.39    | 1.61     |
> | SD-V2.1 | 768     | 2.72 / 6.51       | 2.39     |
> | SD-XL | 1024      | 6.91 / 19.49     | 2.82      |
>
>
> ---
>
> **Q2.** Experiments on diverse model architectures, applications, and datasets.
>
> **A2.** Thanks for the valuable feedback. To address architecture concern, we conduct extra experiments using SD-V2.1, a velocity-parametrized diffusion model; results can be found in A3 of Reveiwer BefU. Below is a summary of the models used, covering a wide range of settings.
>
> Regarding applications and datasets, we respectfully clarify that our choices are **consistent with current standards in the literature**. For examples, DiT (William Peebles and Saining Xie, 2023) and EDM2 (Tero Karras et al., 2024) primarily conduct experiments on ImageNet. In addition to ImageNet, Interval Guidance (Kynkäänniemi et al., 2024) and Autoguidance (Tero Karras et al., 2024) perform qualitative experiments on text-to-image tasks using SD models. Due to limited time and resource constraints, we will explore additional datasets in future work.
>
>
> | Model           | DiT-XL/2         | EDM2-S           | EDM2-XXL         | SD-v1-4         | SD-XL                | SD-V2.1           |
> |-|-|-|-|-|-|-|
> | # Params     | 675 M           | 280 M            | 1.5 B            | 860 M           | 2.6 B               | 865 M             |
> | Sampler          | 250-step DDPM   | 2nd Heun | 2nd Heun | PNDM    | Euler Discrete | PNDM      |
> | Parametrization  | epsilon | x0 | x0 | epsilon | epsilon | velocity |
> | Architecture     | Transformer      | U-Net            | U-Net            | U-Net           | U-Net               | U-Net             |
>
> ---
>
> **Q3.** Comparison with recent guidance enhancements.
>
> **A3.** Thanks for the constructive suggestion. We respectfully note that the proposed REG method has already been compared with SOTA guidance techniques, such as Interval Guidance (Kynkäänniemi et al., 2024) and Autoguidance (Tero Karras et al., 2024), both of which are strong and recent baselines. As shown in Table I and Figure 7, REG is still able to improve upon these methods. We greatly appreciate suggestions from the reviewer on any specific missing guidance methods that we should compare to.
>
> ---
>
> **Q4.** REG implementation details and open-source code.
>
> **A4.** Thanks for the constructive remark. We will open source our code to ensure full reproducibility, and add implementation details in the updated paper.
>
> ---
>
> **Q5.** Other comments: (i) Add intuitive explanation for marginal scaling in Section 3. (ii) Add a summary table of results and architectures in numerical result section. (iii) REG with different sampling schemes and different parametrizations.
>
> **A5.** Thanks for the constructive remarks.
>
> (i) We will update our manuscript accordingly.
>
> (ii) We want to respectfully point out that Table 2 and 3 have already included all numerical results and Table 4 (in supplementary) have summarized model architectures.
>
> (iii) We respectfully note that our experiments already cover a wide range of settings. Please refer to the summary table in A2 for details.

---

### Official Review · Reviewer_BefU · 2025-03-13

**Overall Recommendation:** 5

**Summary:**

This submission focuses on demystifying the classifier-free guidance for diffusion models. CFG has proven to be essential for the success of diffusion models. However, recent literature noted that the guided score function does not correspond to the forward diffusion process. In this work, the authors identify the source of the discrepancy and then introduce guidance for the joint distribution. This results in guidance relying on the expected reward at $x_0$ at every timestep of sampling. Using such formulation directly would be computationally prohibitive, as it would require completing the denoising process to the terminal state, at every timestep. However, under mild assumptions, which can be met in practice, a convenient approximation is proposed, which the authors call “rectified gradient guidance”. REG comes at the cost of computing the diagonal of the Jacobian of the denoiser. The proposed method is supported by strong evidence in toy, controlled scenarios, as well as for state-of-the-art class- and text-conditional image generation diffusion models. In addition, the authors show the standard CFG can be seen as an approximation of REG, and characterize the approximation error.

## update after rebuttal
My assessment remains positive after the rebuttal. The authors engaged in the discussion and provided additional details that were requested.

**Claims And Evidence:**

Both the claims regarding theoretical results, as well as, claims about the effectiveness of the proposed method are supported by appropriate evidence.

**Essential References Not Discussed:**

Nothing that was available at the time of submission. The authors may find [1] interesting if they didn’t know it already,

[1] Pavasovic, Krunoslav Lehman, et al. "Understanding Classifier-Free Guidance: High-Dimensional Theory and Non-Linear Generalizations." arXiv preprint arXiv:2502.07849 (2025).

**Experimental Designs Or Analyses:**

Yes. The synthetic experiments in section 5.1. show that the proposed relaxation has a smaller error than the alternative which is vanilla CFG. Image generation experiments in section 5.2. show that this translates to quantitative and qualitative evaluation for SOTA models. There are no issues.

**Methods And Evaluation Criteria:**

Yes.

**Other Comments Or Suggestions:**

- It would be interesting to include samples with the golden guidance in Figure 2, given that it is already computed.
- In Sec 5.2 none of the models uses v-prediction. Not that I expect a different behavior for such a model, but for an even stronger message it could be included.

**Other Strengths And Weaknesses:**

This is a strong submission.

**Questions For Authors:**

1. L243 right column “assuming that the Jacobian matrix is diagonally dominant” - is diagonal dominance enough? I think that the assumption here is a diagonal Jacobian.
2. On the approximation of eq 22 with eq 21: wouldn’t the efficient vector-jacobian-product be applicable here? This would eliminate the assumption mentioned in the question above.
3. What is the computational overhead of the proposed method compared to vanilla CFG? Despite the use of approximation, I suspect that the additional computational and memory cost is significant for any reasonably sized model.

**Relation To Broader Scientific Literature:**

This submission continues on the path of demystifying, and trying to understand CFG. I agree with the authors that their work complements previous findings in this field. I am inclined to believe that this is the explanation we have been looking for.

**Theoretical Claims:**

I have reviewed the proofs. I focused on understanding what technique the authors used for each of the proofs. I have not thoroughly checked all derivations. I have no reason to believe they are incorrect.

---

> ### Author Rebuttal · Authors · 2025-03-30
>
> Thank you so much for acknowledging our contribution and the constructive feedback.
>
> ---
>
> **Q1.** Reference [1]: Krunoslav Pavasovic et al., arXiv 2025.
>
> **A1.** Thanks for bringing [1] to our attention. We are aware of this work and will include it in our references. As the reviewer kindly noted, this paper was published on arXiv on February 11th, after the ICML submission deadline. [1] shows that while CFG may reduce diversity in low-dimensional settings, it becomes effective in high-dimensional regimes due to a "blessing of dimensionality." The authors identify two phases: an early phase where CFG aids class selection, and a later phase where it has minimal impact. They propose non-linear CFG variants that deactivate in the second phase, improving quality and diversity without extra computational cost.
>
> ---
>
> **Q2.** Include samples with the golden guidance in Figure 2.
>
> **A2.** Thanks for the great question. We respectfully clarify that Figure 2(a) already displays the golden/target samples that we aim to generate, and Figure 2(d) presents the golden guidance $\nabla \log E_t$. To the best of our understanding, these are all the golden cases available. We are more than happy to include any missing visualizations to address this concern.
>
> ---
>
> **Q3.** Extra results on diffusion models using velocity-parametrization.
>
> **A3.** Thanks for the constructive feedback. We conduct extra experiments on SD-V2.1, a velocity-parametrized text-to-image diffusion model. Due to time and resource constraints, we perform a grid search over guidance weights and report the best FID $\downarrow$ and CLIP $\uparrow$ scores for each method in the table below. A more thorough evaluation (e.g., a full FID-CLIP curve) will be included in the final version if accepted.
>
> |        | (CLIP, FID) w/ REG          | (CLIP, FID) w/o REG       |
> |-|-|-|
> | Linear CFG    | (31.62, 27.83)           | (31.40, 28.94)         |
> | Cosine CFG    | (31.48, 23.32)           | (31.72, 24.54)         |
>
>
> ---
>
> **Q4.** Diagonal Jacobian matrix assumption.
>
> **A4.** Thanks for the constructive remark. We agree that a diagonal Jacobian is more precise. We originally refer to a diagonally dominant Jacobian for a writing reason. From Eq. (22) to Eq. (21), three approximations are introduced, as detailed in Lines 220–224 on the right side of the paper, with the Jacobian assumption being the third. The first two approximations are essential—(2) is self-explanatory, and (1) is explained in our response to Q5. Given these, the transition from Eq. (22) to Eq. (21) is already approximate, regardless of whether the Jacobian is diagonal or diagonally dominant.
>
> We will revise the text to refer to a diagonal Jacobian where appropriate.
>
> ---
>
> **Q5.** Apply Vector-Jacobian-Product (VJP) in Eq. (22).
>
> **A5.** Thanks for the great question. In short, the application of VJP requires that the function $\log R_0(\hat{x}_0, y)$ be differentiable with respect to $\hat{x}_0$. However, in current CFG-like frameworks, $R_0(\cdot, y)$ is not arbitrary—it is specifically defined as shown in the second line of Eq. (6). This definition renders $\log R_0(\cdot, y)$ non-differentiable with respect to its first argument (or very consuming to evaluate). Consequently, VJP is not applicable in this case.
>
> Note that to address it, we use chain rule and approximate $\nabla_{\hat{x}_0} \log R_0(\hat{x}_0, y)$ with $\nabla{x_t} \log R_t(x_t, y)$, which can be further simplified via Eq. (7) and (9).
>
> ---
>
> **Q6.** Runtime and memory cost.
>
> **A6.** Thanks for the great question. The tables below summarize runtime and peak memory usage of CFG and REG on a single NVIDIA A40 GPU. Runtime is reported using example batch sizes, while memory is measured with batch size 1 to isolate per-image cost. As expected, REG introduces minor overhead due to the extra gradient computation. We will add these tables in the updated paper.
>
>
> | Model | Resol. | Batch Size | CFG/REG Runtime (sec) | Increase (x) |
> |-|-|-|-|-|
> | EDM2-S | 64    | 8          | 25.96 / 42.99        | 1.66   |
> | DiT-XL/2 | 256 | 8          | 59.79 / 94.23        | 1.58     |
> | EDM2-S | 512   | 8          | 46.14 / 62.87        | 1.36      |
> | EDM2-XXL | 512 | 8          | 49.21 / 92.60        | 1.88      |
> | SD-V1.4 | 512  | 4          | 32.63 / 39.54        | 1.21      |
> | SD-V2.1 | 768  | 4          | 36.55 / 59.76        | 1.64   |
> | SD-XL | 1024   | 2          | 47.48 / 74.52        | 1.57    |
>
> | Model | Resol.   | CFG / REG GPU Peak Mem (GB) | Increase (x) |
> |-|-|-|-|
> | EDM2-S | 64       | 0.87 / 1.49   | 1.71  |
> | DiT-XL/2 | 256    | 4.15 / 5.01    | 1.21   |
> | EDM2-S | 512      | 1.19 / 1.81      | 1.52   |
> | EDM2-XXL | 512    | 4.59 / 7.31   | 1.59    |
> | SD-V1.4 | 512     | 2.73 / 4.39      | 1.61   |
> | SD-V2.1 | 768     | 2.72 / 6.51       | 2.39    |
> | SD-XL | 1024      | 6.91 / 19.49       | 2.82    |

---

> > ### Comment · Reviewer_BefU · 2025-04-03
> >
> > I thank the authors for a very well-organized reply and the additional information provided. Please find my additional questions/comments below:
> >
> > 1. A&Q 2
> >
> > Would it be possible to generate samples with the golden guidance from Figure 2(d)? Or is this exactly what we see in Figure 2(a)?
> >
> > 2. A&Q 5
> >
> > What I meant was using jacobian-vector-product (not VJP) in eq 22, using the last approximation used in your reply. I even implemented it for the toy model provided in the supplementary material, and it seems to work fine.
> >
> > 3. Additional comment
> >
> > As mentioned by other reviewers, it would be great if you could release the code for all the experiments.

---

> > > ### Author Response · Authors · 2025-04-03
> > >
> > > Thank you for acknowledging our responses and the thoughtful follow-up questions.
> > >
> > > ---
> > >
> > > **1. Response to Q&A 2**
> > >
> > > Thank you for the clarification — we now better understand the question. It indeed looks simialr to Figure 2 (a), so we have omitted it since Figure 2 already has 8 columns.
> > >
> > > ---
> > >
> > > **2. Response to Q&A 5**
> > >
> > > Thank you for the clarification, the insightful suggestions, and even trying to implement it in our supplementary toy code. We now understand that the reviewer is referring to the Jacobian-Vector Product (JVP) --- applying JVP to the final line of Eq. (22) after we approximate $\nabla_{\hat{x}_0} \log R_0(\hat{x}_0, y)$ with $\nabla{x_t} \log R_t(x_t, y)$.
> > >
> > > We agree with the reviewer that using JVP is a valid and promising approach here. It can also elminate the need for the diagonal Jacobian assumption. Taking CFG as an example and using our notation, we know $-\sqrt{1-\bar{\alpha}_t}\nabla{x_t} \log R_t(x_t, y)=\epsilon_t(x_t,y,t) - \epsilon_t(x_t,t)$. This suggests that JVP for the last line of Eq. (22) can be written roughly in the following pseduo code:
> > >
> > > ```python
> > > # net(xt, y, t) is the trained conditional noise prediction network
> > > uncond_pred = net(xt, None, t) # unconditional noise prediction without labels
> > > cond_pred = net(xt, y, t) # conditional noise prediction with labels
> > > pred = cond_pred + w * jvp(net, xt, cond_pred - uncond_pred)
> > > ```
> > >
> > > We sincerely appreciate the reviewer’s insightful suggestion, which we had not considered in our original method. In our view, compared to our current implementation, the JVP-based formulation introduces one fewer approximation and therefore has the potential for theoretically even better performance.
> > >
> > > A complete evaluation requires testing JVP on our full experimental setup (i.e., class-conditioned ImageNet and text-to-image tasks), which cannot be completed within the rebuttal period since we need to sweep the FID v.s. IS (or FID v.s. CLIP) curves. We will implement and evaluate the JVP-based variant and will update the manuscript accordingly. Once the updated version is publicly available, we welcome any further feedback or suggestions from the reviewer..
> > >
> > > Finally, we want to thank the reviewer again for this important remark.
> > >
> > > ---
> > >
> > > **3. Additional Comment**
> > >
> > > Thank you so much for the feedback. We plan to release all the code for full reproducibility if the paper is accepted. Additionally, we will incorporate all discussions from the rebuttal period into the updated manuscript. Most importantly, we will implement the JVP, examine it, and add its results.

---

### Decision · Program_Chairs · 2025-05-01

**Decision:**

Accept (poster)

**Comment:**

The paper introduces Rectified Gradient Guidance (REG), a method to correct theoretical flaws in existing conditional diffusion model guidance. It shows that previous methods incorrectly scale marginal distributions and proposes scaling the joint distribution instead, establishing a valid theoretical foundation. REG approximates this corrected optimal solution to make it practical and improve conditional generation quality. Extensive experiments validate REG's superiority, showing consistent performance improvements in synthetic and real-world image generation tasks over existing techniques.

All reviewers feel positive about this paper, mainly due to the theoretical contribution and practical approximation of the proposed method. There are some concerns regarding some experiments details, which have been well addressed in the rebuttal. Please make sure to incorporate additional results and clarifications into the final version to make the paper stronger.